# CooT: Learning to Coordinate In-Context with Coordination Transformers

Huai-Chih Wang [* 1]  Hsiang-Chun Chuang [* 1]  Hsi-Chun Cheng [1]  Dai-Jie Wu [1 2]  Shao-Hua Sun [1 3]

## Abstract

Effective coordination among unfamiliar partners remains a major challenge in multi-agent systems. Existing approaches, such as population-based methods, improve robustness through diversity but often lack mechanisms for efficient adaptation beyond training distribution. Moreover, fine-tuning is impractical in few-shot settings due to its high interaction cost. To address these limitations, we propose Coordination Transformer (CooT 🪐), a framework that leverages in-context learning (ICL) for real-time partner adaptation. Unlike prior ICL approaches that focus on task generalization, CooT is designed to generalize across diverse partner behaviors. Trained on trajectories from behavior-preferring agents, it learns to align actions with partner intentions purely through observation. We evaluate CooT on two challenging multi-agent benchmarks: Overcooked and Google Research Football. Results show that CooT consistently outperforms population-based methods, gradient-based fine-tuning, and Meta-RL baselines, achieving stable and rapid adaptation without parameter updates. Human evaluations also identify CooT as a preferred collaborator, and our ablations confirm its ability to adapt quickly to new partners and remain stable under sudden partner changes, making it reliable for real-world human-AI collaboration. Demos are available at https://coot-project.github.io/coot/.

*Equal contribution  [1]Graduate Institute of Communication Engineering, National Taiwan University (NTU)  [2]University of Utah  [3]NTU Artificial Intelligence Center of Research Excellence (NTU AI-CoRE). Correspondence to: Huai-Chih Wang <d14942005@ntu.edu.tw>, Shao-Hua Sun <shao-huas@ntu.edu.tw>.

*Proceedings of the 43^{rd} International Conference on Machine Learning*, Seoul, South Korea. PMLR 306, 2026. Copyright 2026 by the author(s).

## 1. Introduction

Coordination is fundamental to intelligent behavior, enabling people to achieve shared goals through joint effort. In dynamic environments, such as team sports or traffic navigation, humans adjust their actions based on the behaviors and intentions of others, enabling collaboration. Replicating this adaptive coordination in artificial agents remains a core challenge in multi-agent systems (Cao et al., 2012; Zhang et al., 2021), especially in domains like robotics (Yan et al., 2013; Zhang et al., 2023; Ko et al., 2024), gaming (Matignon et al., 2012), and human-AI interaction (Carroll et al., 2019). This challenge is commonly studied as the ad-hoc teamwork problem (Stone et al., 2010), where agents coordinate with previously unseen partners.

Methods for agents to coordinate with unseen partners have used various strategies. Self-play (SP; Tesauro 1994; Yu et al. 2022a; Hu et al. 2021; Wang et al. 2023) trains agents by having them repeatedly interact with copies of themselves. While effective in simple environments or when coordinating with familiar partners, SP often converges to conventions that break down when interacting with unfamiliar collaborators. Population-based methods (Jaderberg et al., 2017) seek to address this by training agents within diverse populations using strategies such as partner randomization (Hu et al., 2020; Lucas & Allen, 2022), reward shaping (Yu et al., 2023). However, while these approaches promote robustness, they often lack mechanisms for efficient online adaptation, which limits their ability to generalize beyond the training distribution. Fine-tuning policies on new partners (Nekoei et al., 2023) is also proven ineffective, since adaptation often demands thousands or even millions of interaction trajectories to learn effective coordination strategies, thus making few-shot fine-tuning infeasible. To address this, prior works (Ma et al., 2024; Lou et al., 2023) have explored context-based approaches that adapt to new partners by conditioning on recent interactions. While effective in reducing extensive fine-tuning, these methods typically rely on compressed contexts learned during training. Such representations can struggle to capture rich, temporally structured partner behaviors and can fail when test-time behaviors deviate from those seen in training.

Our key insight is to leverage the recent advancements in in-

context learning (ICL; Brown et al., 2020; Wei et al., 2022; Li et al., 2023), enabling a fixed model to adjust its behavior at test time, to efficiently and adaptively coordinate with new partners. Originally demonstrated in language modeling (Brown et al., 2020), ICL has recently been adapted to decision-making learning (Chen et al., 2021; Lee et al., 2023) and algorithm distillation (Kirsch et al., 2023; Laskin et al., 2023). However, applying it to coordination poses distinct challenges. Unlike task-focused settings with different rewards, coordination requires aligning with diverse partner behaviors, often without explicit feedback. Thus, the challenge shifts from generalizing across tasks to generalizing to new partners based on interactions.

To address this challenge, we introduce Coordination Transformer (CooT 🌀), a framework designed for in-context partner adaptation. While prior ICL-inspired methods (Chen et al., 2021; Laskin et al., 2023) that focus on task generalization, CooT leverages in-context learning to generalize across diverse partner behaviors. CooT predicts actions that best align with the observed partner's behavior to maximize collaboration effectiveness. To achieve this, we train CooT on trajectories collected from interactions between pairs of agents: a behavior-preferring agent and its corresponding best-response counterpart. Behavior-preferring agents operate according to hidden event-based reward functions. These event-based preferences induce diverse behaviors within identical environments. By observing how a best-response agent adapts its actions to behavior-preferring partners, CooT learns context-driven strategies that approximate best-response behavior, enabling effective coordination with unseen partners.

We extensively evaluate CooT on two multi-agent benchmarks: Overcooked (Carroll et al., 2019), which requires precise two-agent coordination, and Google Research Football (Kurach et al., 2020), which involves coordination in larger teams with more than two partners. Across both environments, we compare CooT against a broad set of representative baselines, including population-based methods such as HSP (Yu et al., 2023) and MEP (Zhao et al., 2023), gradient-based online fine-tuning approaches, context-conditioned meta-RL methods designed for rapid partner-aware coordination, e.g., PEARL (Rakelly et al., 2019), and PACE (Ma et al., 2024). The results demonstrate that CooT consistently achieves strong performance relative to all baselines across a wide range of coordination tasks involving unseen RL-based agents. Beyond these objective performance metrics, human partners perceive CooT as a more adaptive and reliable collaborator. Our analyses show that this advantage comes from the ability to refine coordination strategies in a few-shot manner, achieving effective alignment within only a handful of interactions. This in-context capability also ensures robustness against non-stationary behaviors, where CooT recalibrates to sudden partner shifts without requiring parameter updates.

The core contribution of CooT is a new way to think about how agents work together: we treat partner adaptation as a learning task. By observing a partner's past behaviors, CooT identifies their style and automatically chooses the best response. This allows for fast and flexible coordination that works instantly, without the need to fine-tune the model when a partner's behavior changes.

## 2. Related Work

**Learning to Coordinate.** A fundamental challenge in multi-agent reinforcement learning is learning strategies that remain effective when paired with unseen partners, a setting known as ad-hoc teamwork (AHT; Stone et al. 2010). A collective line of work, referred to as zero-shot coordination (ZSC; Hu et al. 2020; Treutlein et al. 2021; Hu et al. 2021), focuses on how agents can coordinate in a zero-shot manner, i.e., without adapting to task-specific interaction during inference.

Self-play (SP; Tesauro 1994; Yu et al. 2022a; Wang et al. 2023) is a popular baseline that can produce strong agents in simple environments. However, SP policies often converge to *different conventions*, leading to severe failures in complex environments or when paired with partners with different preferences (Carroll et al., 2019). To improve robustness, population-based methods (Jaderberg et al., 2017; Long* et al., 2020; Hu et al., 2020; Strouse et al., 2021) expose agents to diverse partners, with further extensions based on entropy maximization (Zhao et al., 2023; Lupu et al., 2021), reward shaping (Lucas & Allen, 2022), structured population groups (Xue et al., 2024), and hidden-utility biases (Yu et al., 2023). Recent work also explored training agents across large sets of procedurally generated environments (Jha et al., 2025), encouraging general cooperative behaviors that transfer to new partners and tasks. In parallel, the rise of large language models (LLMs) has led to LLM-assisted coordination (Zhang et al., 2024; Li et al., 2025), which leverage natural-language reasoning to improve adaptivity. Beyond cooperative teamwork, related work includes competitive or socially structured settings, see Appendix E.

Despite these advances in ZSC and AHT, a key limitation remains: most approaches rely on large amounts of training interaction to cover diverse partner behaviors. To address this problem, few-shot context-based coordination has been proposed, aiming to train agents in a meta-RL manner to adapt to novel partners with minimal experience. Methods such as PACE (Ma et al., 2024) and PECAN (Lou et al., 2023) infer partner behavior from limited trajectories, while LIAM (Papoudakis et al., 2021) learns to adapt through partner representations from local interactions. However, these methods remain limited. PACE relies on matching unseen

partners to training identities, which can break down when partners exhibit novel behaviors. PECAN largely models partner skill levels, neglecting behavioral and strategic variation, while LIAM's latent trajectory encoding can introduce information loss, limiting effective adaptation. Building on this line of work, our method CooT leverages the in-context learning capabilities of transformers to achieve effective few-shot coordination with unseen partners. Unlike previous few-shot approaches (*e.g.*, PACE, LIAM), CooT learns from full interaction histories to directly predict actions that best complement the partner's behaviors, enabling efficient test-time adaptation without parameter updates.

**In-Context Reinforcement Learning.** In-context learning (ICL; Brown et al. 2020; Wei et al. 2022; Li et al. 2023) enables models to adapt at inference by conditioning on examples rather than gradient updates. Beyond its success in language and vision (Brown et al., 2020; Yu et al., 2022b), recent studies extend ICL to reinforcement learning (Chen et al., 2021; Jing et al., 2023; Laskin et al., 2023; Lee et al., 2023), showing applications in offline RL, online decision-making, opponent modeling, and meta-RL. Prior works suggest that transformers may implicitly implement optimization-like procedures (Von Oswald et al., 2023), with trajectories serving as contextual examples for learning algorithms or posterior sampling (Laskin et al., 2023; Lee et al., 2023; Jing et al., 2023). This perspective is closely related to transfer and meta-learning in RL, which aim to generalize across tasks using prior experience (Duan et al., 2016; Finn et al., 2017; Rakelly et al., 2019; Tobin et al., 2017; Zhang et al., 2020; Xing et al., 2021; Vuorio et al., 2019; Nam et al., 2022). However, unlike traditional transfer learning approaches that rely on explicit parameter updates for adaptation, in-context RL achieves implicit transfer by conditioning a fixed model on recent interaction histories. Recent work extends this paradigm to multi-agent settings. Transformers have been shown to deploy best-response policies across co-players in mixed-motive settings (Weis et al., 2026), and TAGET (Zhang et al., 2025) applies offline goal-based decision transformers to ad-hoc teamwork but requires subgoal supervision and does not support online adaptation. Partner modeling has also been shown to emerge implicitly in recurrent agents (Mon-Williams et al., 2025). Building on these, CooT provides a practical framework that leverages cross-episode context for test-time partner adaptation without parameter updates, targeting pure cooperative coordination under shared rewards.

## 3. Preliminary

### 3.1. Hidden-Utility Markov Game

We formulate the in-context coordination problem as a Hidden-Utility Markov Game (HU-MG), inspired by the framework introduced in Hidden-Utility Self-Play (Yu et al.,

2023). The HU-MG builds on the two-agent decentralized Markov game (Daniel S Bernstein, 2002), defined as the tuple $(\mathcal{S}, \mathcal{A}, \mathcal{T}, \mathcal{R}_t)$, where $\mathcal{S}$ denotes the state space, $\mathcal{A} = \mathcal{A}^i \times \mathcal{A}^w$ is the joint action space, $\mathcal{T} : \mathcal{S} \times \mathcal{A}$ represents the transition function, and $\mathcal{R}_t$ the shared global reward. Both agents, $\pi_i$ and $\pi_w$, aim to maximize $\mathcal{R}_t$.

A HU-MG is instead defined as $(\mathcal{S}, \mathcal{A}, \mathcal{T}, \mathcal{R}_t, \mathcal{R}_w)$, where $\mathcal{R}_w$ is a hidden reward space reward functions $r^w \sim \mathcal{R}_w$. The hidden reward functions $r^w$, observable only by $\pi_w$, are formulated as linear combinations over event features, inducing distinctive behavioral preferences or conventions. Note that $\pi_w$ is the partner policy $\pi_i^p$ introduced in Section 4, but we adopt the new notation to better distinguish partner indices in the dataset setting.

### 3.2. In-Context Learning with Decision-Pretrained Transformers

In-context learning refers to a model's ability to generalize by conditioning predictions on contextual examples at inference time. A Decision-Pretrained Transformer (DPT) (Lee et al., 2023) trains on sequences of $(s, a, r)$ triplets, representing states, actions, and returns. In inference, it autoregressively predicts the optimal action for a given state, conditioned on contextual examples. This reframes reinforcement learning as sequence modeling, where task identity is inferred from context. However, standard DPT assumes access to extrinsic rewards and explicit task objectives, limiting applicability in coordination settings with diverse and implicit partner preferences.

## 4. Method

We propose Coordination Transformer (CooT 🖥️), a framework that adapts to unseen partners, conditioning on interactions. An overview of CooT is illustrated in Figure 1. Our approach involves training a transformer on trajectories from diverse partners and their corresponding **best responses**, *i.e.*, the behavior that yields the highest return given the behaviors of its partners. We begin by formalizing coordination as a Hidden-Utility Markov Game (HU-MG) in Section 4.1, where agents behave according to predefined hidden reward functions. In Section 4.2, we describe how we generate a diverse dataset of interaction histories. In Section 4.3, we explain how CooT is trained to predict optimal actions based on its partner's behavior. Finally, in Section 4.4, we present our protocol for evaluation.

### 4.1. From Task to Coordination Generalization

Recent advances in ICRL enable sequence models to generalize across tasks. However, they do not address the challenge of ad-hoc teamwork, more specifically, adapting to unseen partners with hidden preferences under the same

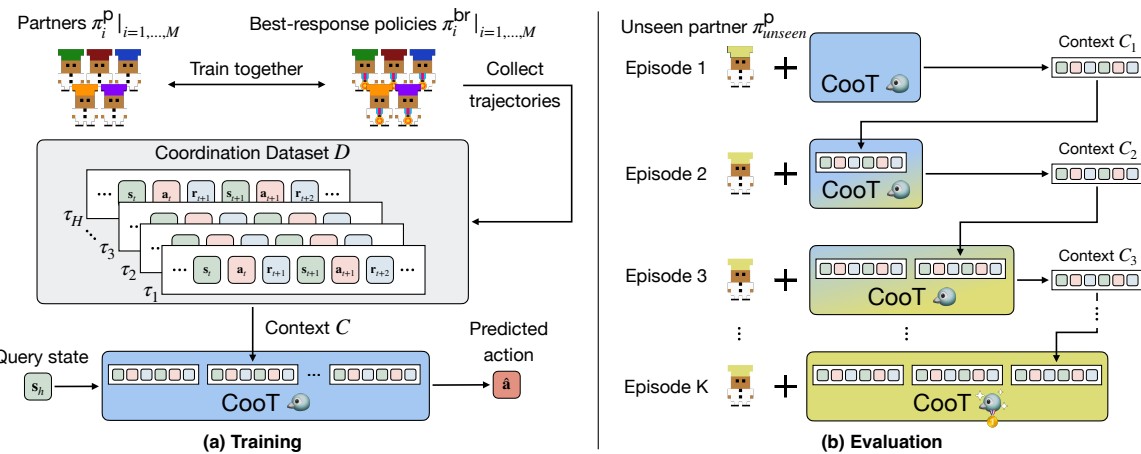

*Figure 1.* **Overview of CooT. (a) Training.** We generate a dataset $D$ of trajectories between behavior-preferring agents and their best-response (BR) policies. For each training data, CooT receives query states $s_h$ and context $C$ of past interactions, and learns to predict action approximating the BR action $\hat{a}$. **(b) Evaluation.** At test time, CooT coordinates with unseen partners by continually updating its context from recent episodes. This mechanism allows few-shot generalization by adapting to the partner through context alone.

task. We reframe this challenge as a multi-player Hidden-Utility Markov Game (HU-MG), where behaviors of agents are driven by hidden reward functions that are unobservable to other agents. We refer to such agents as **behavior-preferring** agents, since their preferences are revealed only through behavior. These hidden utilities induce diverse strategies, even when the environment remains fixed. In this setting, adapting to behavior-preferring agents requires inferring their underlying preferences through their actions.

The novelty of CooT lies in the way it defines the learning target. While our architecture builds on the Decision-Pretrained Transformer (Lee et al., 2023), we shift the objective from general task-solving to partner-specific best-response. We treat each unique partner behavior as a distinct "task" to be solved. By observing cross-episode histories, CooT identifies the behavioral pattern of the partner and predicts the corresponding best response, adapting to the partner rather than the environment. This approach allows CooT to achieve rapid, gradient-free coordination by mapping observed partner styles directly to optimal collaborative strategies.

### 4.2. Dataset Generation

Before training, we first construct a dataset $D$ containing interactions between behavior-preferring agents and their corresponding best responses. Our data generation process is grounded in the HU-MG framework as described in Section 3.1, where coordination challenges arise from hidden reward differences.

Following prior work on population-based coordination (HSP; Yu et al. 2023), we simulate hidden preferences by first defining a set of discrete environmental events. We then,

as illustrated in Algorithm 1, construct different hidden reward functions using linear combinations of environmental events to form a reward space $\mathcal{R}_w$. For each hidden reward function $r_i^w \sim \mathcal{R}_w$, we jointly train a behavior-preferring partner policy $\pi_i^{\text{p}}$, driven by $r_i^w$, and its best-response policy $\pi_i^{\text{br}}$ using Multi-Agent Proximal Policy Optimization (MAPPO; Yu et al. 2022a). Each trained pair $(\pi_i^{\text{p}}, \pi_i^{\text{br}})$ is added to the initial pool $\Pi_0$ afterward.

To ensure diversity among training partners, we compute an event-based diversity score $d_i$ for each best-response policy $\pi_i^{\text{br}}$, and select the top-$N$ diverse pairs to form the training pool $\Pi_{\text{train}}$. For each selected pair $(\pi_i^{\text{p}}, \pi_i^{\text{br}})$, we collect $T$ trajectories to construct a fixed-length context $C$.

From each context, we construct tuples $(s_h, C, \hat{a})$, where $\hat{a} = \pi_i^{br}(s_h)$ is the best-response action. The query state $s_h$ is sampled independently from the context trajectories and uniformly from the offline dataset $\mathcal{D}$, preventing simple trajectory continuation and forcing the model to infer partner-specific behavior from context.

### 4.3. Training

Given the dataset of context–query–action tuples, we train the Coordination Transformer to perform best-response prediction via in-context learning. At each training step, the model receives query states $s_h$ and a context window $C$, consisting of past interactions between a behavior-preferring agent and its best response. The transformer then predicts the best-response action distribution $\hat{p}_h(\cdot) = M_\theta(\cdot \mid s_h, C)$.

The model is trained to minimize the cross-entropy loss: $\mathcal{L} = -\log \hat{p}_h(\hat{a})$ where $\hat{a} = \pi_i^{\text{br}}(s_h)$ is the ground-truth best-response action given the query states. This objective

encourages the model to map partner behaviors to effective responses, supporting generalization through in-context adaptation to unseen collaborators. Further training details are provided in Appendix B.2.

### 4.4. Online Deployment

Before deployment, we first need to generate an evaluation population. As illustrated in Algorithm 2, we extract behavioral features $\phi_i$ from trajectories generated by each $\pi_i^{\text{br}}$ and compute the population diversity. To efficiently select a diverse evaluation set, we then apply Determinantal Point Process sampling (Kulesza et al., 2012) over **K**, a similarity matrix of behavioral feature set $\{\phi_i\}$, resulting in a behaviorally diverse evaluation set $\Pi_{\text{eval}} \subset \Pi_0$. Details on the similarity matrix construction and the full evaluation pipeline are provided in Appendix B.6.

COOT is deployed with unseen behavior-preferring partners $\pi_{unseen}^{\text{p}} \sim \Pi_{\text{eval}}$ over multiple episodes of length $Z$ at test time. To isolate COOT's few-shot adaptation capability, we adopt a strict no-prior protocol: we initialize a fixed-length context $C$ consisting of $T$ empty trajectories before deployment begins. At each timestep $t$ in an episode of length $Z$, the agent observes context $C$, current state $s_t$, and a short history of prior states (*e.g.*, $\{s_{t-n}, \ldots, s_t\}$), which together form the query state $s_h$. Based on this input, COOT predicts an action distribution $\hat{p}_t(\cdot) = M_\theta(\cdot \mid s_h, C)$ and samples an action $a_t \sim \hat{p}_t$.

After executing action $a_t$ and observing reward $r_t$, the transition $(s_t, a_t, r_t)$ is recorded. Once the episode ends, the resulting trajectory, which is denoted as $\tau = (s_t, a_t, r_t)_{t=1}^Z$, is appended to the context buffer and pushes out the oldest trajectories. Repeating this process across $E$ episodes allows COOT to refine its coordination strategy through in-context adaptation. Without parameter updates, it continuously adapts to partner behaviors using recent interactions, demonstrating few-shot generalization in multi-agent coordination.

## 5. Experiments

### 5.1. Evaluation Setup

**Environments.** We evaluate COOT on two distinct multi-agent benchmarks: Overcooked (Lauffer et al., 2023), a discrete grid-world task requiring tight coordination and task allocation; and Google Research Football (Kurach et al., 2020), a continuous, high-dimensional environment that tests coordination in a more complex, multi-agent setting.

- **Overcooked.** Our primary evaluation environment, where two agents prepare and deliver soups within a fixed time. Agents can move, wait, or interact with ingredients (e.g., onions, tomatoes) and objects (e.g.,

pots, plates, dispensers), receiving a sparse reward of 20 for each successful delivery. To assess generalization, we test across five layouts, as shown in figure 3: *Asymm. Adv.*, *Bothway Coord.*, *Blocked Coord.*, *Co-ord. Ring*, and a multi-recipe variant of it. Each layout introduces distinct coordination challenges through resource distribution and spatial constraints, requiring different levels of coordination.

- **Google Research Football (GRF).** To further examine the scalability of COOT, we use the *3 vs 1 with Keeper* scenario in GRF, as shown in figure 4. In this setting, three attackers, denoted as $P_0$, $P_1$, and $P_2$, must coordinate their movements and bypass the defender to score. To increase coordination difficulty, we define two sets of discrete events for $P_0$ and $P_2$, and generate biased rewards through combinations of these events. We then collect trajectories of $P_1$ acting as the best response to each pair of partners ($P_0$, $P_2$) and use these to train CooT. The performance is measured by the goal rate, defined as the number of successful goals achieved within each episode (further details in Appendix A.2).

**CooT.** COOT uses a GPT-2 transformer architecture (Radford et al., 2019) as its backbone following DPT (Lee et al., 2023). The backbone is trained via supervised learning on trajectories of best-response agents. COOT is trained on a dataset of trajectories collected from 36 best-response policies, each trained against a corresponding behavior-preferring agent drawn directly from the HSP training population. Specifically, only the best-response trajectories are used for training. At each timestep, the query state $s_{\text{query}}$ is appended as the final token to the context sequence $[\tau_1, \tau_2, \ldots, \tau_T, s_{\text{query}}]$. We use a context window of 5 episodes and employ context masking during training. The effect of context length studied in our ablations (Section 5.5). Full architectural and hyperparameter details are provided in Appendix B.

**Baselines.** We compare COOT to a set of coordination methods, ranging from naive imitation and population-based to context-based few-shot frameworks.

- **Behavior Cloning (BC).** Supervised imitation of best-response demonstrations. We consider both a feed-forward BC policy and a recurrent variant (BC-RNN), which conditions on within-episode interaction history through hidden states. This baseline tests whether supervised imitation alone is sufficient for partner generalization.

- **Population-based Baselines.** We compare against Hidden-Utility Self-Play (HSP; Yu et al. 2023) and Maximum Entropy Population (MEP; Zhao et al. 2023), two strong population-based RL baselines for

ad-hoc teamwork that improve partner generalization through behavioral diversity. Both methods first construct a diverse population of partners, and then train an RL policy to coordinate effectively across that population. HSP induces diversity through varied hidden utility functions, whereas MEP promotes diversity via population entropy regularization.

Both HSP and MEP are implemented as recurrent policies, allowing them to condition on past interactions through hidden states during coordination.

- **HSP-fine-tuning (HSP-ft).** We apply gradient updates on interaction data during evaluation, yielding a fine-tuning baseline called HSP-ft. Included to test whether few-shot improvement can be obtained via gradient updates rather than in-context adaptation.

- **HSP-meta.** For meta-RL context-based comparison, we extend HSP with a trajectory encoder inspired by PEARL (Rakelly et al., 2019), which encodes recent episodes to a latent context appended to the observation. HSP-meta is included to test whether contextual embeddings alone can improve test-time adaptation in an ad-hoc teamwork setting.

- **PACE.** We include another context-based baseline from PACE (Ma et al., 2024), which not only uses an encoder to generate latent context, but also incorporates a peer identifier trained to predict the identity of training agents. Further details in Appendix F.1).

We further include comparisons to broader coordination baselines, including the generative adaptation method GAMMA (Liang et al., 2024) and the policy-selection method PLASTIC (Barrett, 2015), in Appendix F. Full architectural details are provided in Table 12.

**Pipeline.** Following the evaluation protocol introduced by (Wang et al., 2024), which measures ad-hoc teamwork with unseen partners, we evaluate each layout with a fixed set of partner policies. Specifically, we use 10 evaluation partners for all Overcooked layouts and GRF *3 vs 1 with Keeper*, except for the Overcooked *Coord. Ring Multi-recipe*, where 15 partners are used to account for the larger diversity of plausible partner strategies. Evaluation partners are policies trained under diverse hidden reward functions, and a representative subset is selected using Best Response Diversity (BR-div), defined as the determinant of the similarity matrix of their best responses. For each of the evaluation partners, we run 50 consecutive episodes, where COOT continuously adapts by appending trajectories to its context across episodes. Further details, such as the full formulation of BR-div, are provided in Appendix B.6, and the robustness checks, including the overlap between training and evaluation populations, and the evaluation partner set sensitivity analyses, are in Appendix B.7.

Unlike the original protocol, which evaluates across multiple training checkpoints, we evaluate only fully converged checkpoints to remove variability from partially trained behaviors and isolate final performance.

**Evaluation Metrics.** We report Best Response Proximity (BR-prox) and average return. BR-prox (Wang et al., 2024) quantifies how close an agent comes to the best-response performance when paired with a given evaluation partner. More precisely, BR-prox is defined as the ratio between the agent's return and the return achieved by the best-response policy with that partner. A higher ratio indicates stronger coordination and generalization. To avoid degenerate cases, we exclude partners whose best-response return is zero.

## 5.2. Coordination Performance Across Benchmarks

### 5.2.1. AD-HOC COORDINATION IN OVERCOOKED

COOT delivers consistently strong results across diverse Overcooked layouts, with its advantage becoming more pronounced as coordination demands increase. As shown in Table 1, COOT is competitive in *Coord. Ring* and shows clearer improvements in *Coord. Ring Multi-recipe*, where agents must resolve concurrent goals and coordinate more tightly. Layout structure primarily determines how much coordination is required. In layouts where agents can act independently (e.g., *Asymm. Adv.*), simple methods such as BC or population-based RL are often sufficient, and COOT performs comparably. In contrast, coordination-heavy layouts introduce partner-dependent behavior, where conditioning on interaction history becomes useful. COOT leverages this context to infer partner behavior and select appropriate responses. This follows directly from its training objective: predicting best-response actions conditioned on interaction context. At test time, COOT adapts using recent trajectories without explicit partner modeling or gradient updates.

BC and BC-RNN perform reasonably in layouts where agents operate in separated zones, and collisions are avoided, such as *Asymm. Adv.* and *Bothway Coord.*, but degrade in coordination-heavy layouts. Although BC-RNN summarizes within-episode history, it lacks cross-episode context for inferring partner-specific preferences, whereas COOT conditions on recent interaction trajectories and predicts partner-specific responses. This difference becomes increasingly important in coordination-heavy layouts, where adapting to a specific partner's behavior is critical for effective teamwork (Appendix G).

MEP and HSP perform well in simpler settings like *Asymm. Adv.* and compact layouts such as *Coord. Ring*, where interaction is spatially constrained. Yet, as population-based methods, they rely heavily on training partners. In more complex structures (e.g., *Coord. Ring Multi-recipe*), this limited partner pool may fail to capture behaviors of out-of-

*Table 1.* **Benchmark results: COOT outperforms baselines in coordination-heavy layouts.** We report the average return (↑) and BR-prox (↑), both averaged across different layouts. For each method, we run three training seeds. Each seed is evaluated with 50 rollouts, and the per-seed averages are used to compute the final mean ± std reported in the table. We bold results that lie within the best method's standard deviation range for each layout. COOT maintains strong performance across all settings, with clear advantages in coordination-heavy layouts such as *Coord. Ring Multi-recipe* and *Counter Circ.*

| Layout | Coord. Ring | | Coord. Ring Multi-recipe | | Counter Circ. | |
|---|---|---|---|---|---|---|
| | Return | BR-prox | Return | BR-prox | Return | BR-prox |
| BC | 26.24±1.80 | 0.31±0.02 | 8.97±0.49 | 0.10±0.01 | 10.79±5.33 | 0.11±0.06 |
| BC-RNN | 28.07±1.70 | 0.33±0.03 | 21.98±0.19 | 0.25±0.02 | 15.15±0.83 | 0.14±0.04 |
| MEP | **40.30±3.45** | **0.47±0.04** | 16.64±1.16 | 0.19±0.02 | 1.89±0.41 | 0.02±0.00 |
| HSP | **41.10±10.03** | **0.49±0.10** | 29.35±3.77 | 0.33±0.04 | 21.37±1.72 | 0.23±0.03 |
| HSP-ft | **41.30±9.85** | **0.49±0.10** | 29.24±3.75 | 0.33±0.04 | 21.71±1.71 | 0.22±0.02 |
| HSP-meta | 29.84±3.92 | 0.35±0.04 | 30.21±1.37 | 0.34±0.02 | 3.28±0.21 | 0.03±0.00 |
| PACE | **33.94±3.21** | **0.40±0.03** | 4.43±9.02 | 0.05±0.10 | 2.41±0.28 | 0.02±0.01 |
| COOT (Ours) | **38.30±3.71** | **0.47±0.06** | **45.96±3.99** | **0.50±0.04** | **28.28±2.32** | **0.30±0.03** |
| Layout | Asymm. Adv. | | Bothway Coord. | | **Overall** | |
| | Return | BR-prox | Return | BR-prox | Return | BR-prox |
| BC | 108.83±6.13 | 0.53±0.03 | 98.99±1.30 | 0.94±0.01 | 50.76±2.02 | 0.40±0.02 |
| BC-RNN | 105.52±4.20 | 0.51±0.02 | 93.59±1.22 | 0.91±0.01 | 52.86±1.02 | 0.42±0.01 |
| MEP | 127.44±5.66 | 0.61±0.03 | 22.76±5.59 | 0.20±0.05 | 41.81±0.79 | 0.30±0.00 |
| HSP | **134.01±2.19** | **0.63±0.02** | 54.99±3.56 | 0.53±0.03 | 56.16±2.38 | 0.44±0.02 |
| HSP-ft | **133.59±2.34** | **0.63±0.02** | 55.81±2.71 | 0.54±0.02 | 56.33±2.48 | 0.44±0.02 |
| HSP-meta | 113.16±12.72 | 0.54±0.06 | 20.44±4.59 | 0.20±0.05 | 40.19±2.28 | 0.29±0.01 |
| PACE | 124.06±5.45 | 0.59±0.02 | 14.90±11.68 | 0.15±0.12 | 35.95±1.35 | 0.24±0.00 |
| COOT (Ours) | 129.48±9.34 | **0.62±0.05** | **101.93±1.00** | **0.96±0.01** | **68.79±2.33** | **0.57±0.02** |

*Table 2.* **Goal rate on *3-vs-1 with keeper*.** To evaluate coordination beyond two-player setups, we test each method by pairing it with 10 distinct unseen partner pairs in *3-vs-1 with keeper*. The goal rate is defined as goals achieved within 200 steps. Results show mean ± std over 3 seeds, with 50 episodes per seed.

| Method | Goal Rate (↑) |
|---|---|
| BC | 1.97±0.29 |
| MEP | 1.11±0.08 |
| HSP | 1.46±0.31 |
| HSP-ft | 1.52±0.37 |
| CooT (Ours) | **2.50±0.59** |

distribution partners, leading to poor generalization. Unlike HSP and MEP, which train one recurrent policy over a partner population, COOT predicts actions conditioned on recent interaction context. Since COOT uses trajectories from the same HSP-generated partner population, the gains mainly reflect the learning paradigm rather than access to stronger partners. Notably, this comparison is conservative in favor of HSP and MEP, as both are trained with handcrafted dense rewards, whereas COOT relies solely on sparse environmental rewards (Table 26).

HSP-ft further highlights the limitation of gradient-based adaptation. Its performance is highly sensitive to learning rates and shows little improvement across episodes (see Table 17, Appendix C.3). This degradation underscores the importance of in-context adaptation in few-shot settings.

Surprisingly, HSP-meta and PACE both underperform vanilla HSP. To verify this is not due to suboptimal hy-

perparameters, we conduct a systematic hyperparameter search for HSP-meta and find that the best configuration still fails to consistently outperform vanilla HSP across layouts (Appendix B.3). We hypothesize the underperformance stems from three compounding factors: the PEARL-style reconstruction loss favors all trajectory features equally, failing to capture coordination-relevant behavioral patterns; the resulting noisy latents can destabilize policy training; and at test time, the encoder may produce misleading representations for unseen partners. PACE further compounds these issues by routing the trajectory embedding through an additional peer-identifier that predicts partner behavioral identity. Jointly training all three components amplifies instability, consistent with PACE's lower performance relative to HSP-meta. In contrast, COOT conditions directly on raw interaction histories through a transformer, avoiding auxiliary reconstruction losses and partner identity prediction. These results suggest that compressing interaction history into fixed latent representations introduces failure modes that are difficult to overcome.

### 5.2.2. GENERALIZATION TO HIGHER-DIMENSIONAL: GRF

To examine whether COOT generalizes beyond the discrete grid-world of Overcooked, we evaluate it on GRF, which features continuous high-dimensional observations and longer decision horizons. While GRF involves only one additional agent (three total), it presents a meaningfully different challenge from Overcooked in terms of observation complexity and action space. Given the substantially higher computa-

*Table 3.* **Quantitative results of human study.** We report mean return (↑) and participant ratings of collaboration (↑), adaptivity (↑), and best agent (↑), which counts how often a method is selected as the best by participants. CooT achieves the highest scores across all metrics.

| Method | Return | Collab. | Adapt. | Best Agent |
|---|---|---|---|---|
| BC | 51.0±3.8 | 2.0 | 1.8 | 5 |
| MEP | 53.0±5.1 | 3.4 | 2.9 | 8 |
| HSP | 40.5±5.2 | 2.7 | 2.2 | 5 |
| CooT (Ours) | **63.5±9.5** | **4.0** | **3.1** | **18** |

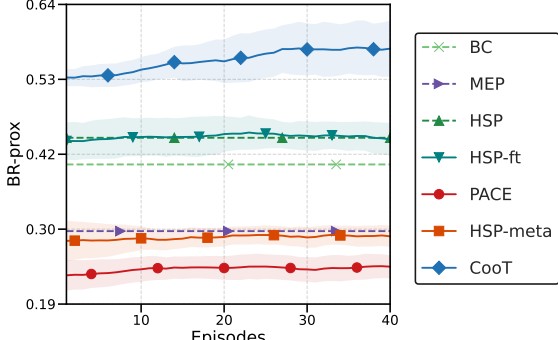

*Figure 2.* **Adaptation performance over episodes.** As more trajectories are observed, CooT steadily improves its coordination strategy, highlighting the advantage of context-based adaptation. Adaptation performance across all five layouts is in Figure 7

tional cost of GRF, we evaluate CooT against a representative subset of strong baselines (BC, MEP, and HSP).

Results in Table 2 show that CooT maintains strong performance in this more complex environment, suggesting that its context-based adaptation is not limited to discrete grid-world settings. We acknowledge that scaling to substantially larger teams remains an open direction for future work.

### 5.3. Human Study

While prior experiments evaluate coordination with policy-based partners, real-world interactions involve humans whose behaviors are more diverse and adaptive. To assess whether CooT 's advantages extend to such settings, we conduct a user study in the Overcooked *Coord. Ring* layout.

Our study passes institutional IRB review and involves 36 participants. Each participant completes 32 rounds across four blocks. To minimize fatigue while maintaining focused comparison, we evaluate four representative agents: CooT, HSP, MEP, and BC. Within each block, participants are paired **consistently** with a single agent for 8 consecutive rounds, followed by subjective ratings and open-ended feedback. Agent order is randomized and double-blind to eliminate ordering effects. Full study details, survey questions, and inference-time comparisons are provided in Appendix B.5 and Appendix D.

**Quantitative Results.** As shown in Table 3, CooT achieves the highest mean return among all methods, outperforming each baseline by a clear margin. Consistent with this objective performance, human evaluations indicate a strong preference for CooT: It receives the highest ratings for collaboration and adaptivity, and is selected as the most preferred partner by 50% of participants (18/36).

**Qualitative Analysis.** To better understand the behavioral factors underlying the observed human preferences, we analyze all $N = 144$ open-ended responses from the human study. Feedback is analyzed with respect to two themes, *Explicit Adaptation* and *Negative Behaviors*, using Gemini 2.5 Flash (Gemini Team, Google, 2025) with manual verification . As shown in Table 21 in Appendix D.3, CooT

is most frequently recognized for explicitly adaptive behaviors, receiving substantially more mentions ($N = 9$) than any baseline. Participants frequently note that CooT improves its coordination over the course of interaction, indicating that these adaptive behaviors are clearly perceived by human partners.

**Discussion.** The human study confirms that CooT successfully generalizes to real-world collaborators. Both quantitative scores and qualitative feedback point to its observable **adaptive capabilities** as a primary strength.

### 5.4. Analysis of Adaptation

Building on the adaptive behaviors observed in the human study, We provide a detailed analysis of CooT's in-context adaptation. Specifically, we focus on two key properties: (i) the efficiency of few-shot adaptation to novel partners, and (ii) robustness under non-stationary partner strategy shifts.

#### 5.4.1. FEW-SHOT ADAPTATION

We first measure how CooT adapts over time when paired with previously unseen partners. Figure 2 reports averages across five Overcooked layouts, each evaluated with 10–15 partners over 50 episodes.

CooT improves within the first 15 episodes and sustains its performance over subsequent episodes, showing that only a handful of trajectories are sufficient to align with a new partner. In contrast, baseline methods show limited or inconsistent adaptation. PACE exhibits only marginal improvement over time, suggesting that its adaptation mechanism is less effective in leveraging a small number of partner trajectories. HSP-ft shows mild fluctuations with no clear upward trend, while HSP-meta adapts more slowly and converges to a lower performance level. Overall, these results highlight CooT 's ability to achieve effective few-shot adaptation under unseen partner behaviors.

*Table 4.* **Adaptation speed after partner switching.** The number of episodes CoOT takes to recover performance after a partner policy change. Recovery is defined as reaching the full non-switching return of the new partner policy. Partner styles are categorized as *Active*, *Still*, and *Dish-averse*, reflecting different biased reward.

| Partner Pair (styles) | Episodes ($\downarrow$) $P_A \rightarrow P_B$ | Episodes ($\downarrow$) $P_B \rightarrow P_A$ |
|---|---|---|
| (Active, Still) | 6 | 5 |
| (Active, Dish-averse) | 6 | 2 |
| (Still, Dish-averse) | 2 | 1 |

### 5.4.2. ADAPTATION TO NON-STATIONARY PARTNERS

Human partners often change strategies mid-task. To test robustness to such non-stationarity, we conduct a controlled partner-switch experiment in the Overcooked *Coord. Ring* layout: CoOT first plays for a few episodes with one HSP biased partner ($P_A$), accumulates context, and is then suddenly paired with a different biased partner ($P_B$) that exhibits a distinct preference. We consider three partner preferences: *Active*, which is penalized for remaining on the same tile; *Still*, which is rewarded for remaining on the same tile; and *Dish-averse*, which receives a large penalty for picking up dishes from the dispenser. We measure the number of episodes it takes for CoOT to reach the non-switching return of the new partner policy, and repeat the procedure in both directions ($P_A \rightarrow P_B$ and $P_B \rightarrow P_A$).

Results in Table 4 show that CoOT adapts within at most six episodes, with an average of 3.67 episodes. This indicates that CoOT can rapidly recalibrate its coordination strategy when faced with abrupt partner changes.

**Discussion.** These results show that CoOT can both adapt quickly to new partners and remain stable under sudden partner changes, two properties that are essential for real-world coordination. We attribute this robust performance to our context design.

### 5.5. Ablation Study

To better understand which context design aspects contribute to these gains, we conduct a series of ablation studies to evaluate the impact of various training-phase configurations and dataset scales on the performance of CoOT. The details of ablations are reported in Appendix C.

**Context Length.** A key design question for in-context adaptation is how much past interaction history to retain for coordination. We vary the number of episodes in the context buffer and find that longer histories improve coordination up to 5 episodes, after which performance degrades, highlighting a trade-off between sufficient evidence and stale or irrelevant history (Table 15, Appendix C). We attribute this to outdated information: as CoOT continuously adapts to its

partner, earlier episodes become less representative of the partner's current behavior, weakening the influence of more relevant recent interactions. We employ context masking during training to encourage reliance on recent context, yet performance still degrades beyond 5 episodes, highlighting a fundamental trade-off between sufficient evidence and outdated history.

**Data Augmentation.** We explore temporal shuffling as data augmentation to improve robustness to interaction history variation. However, not all temporal shuffling forms are effective for coordination. We compare: (1) **Step-wise shuffling**, which randomly permutes all transitions in the context, and (2) **Chunk-wise shuffling**, which preserves short-term temporal structure by shuffling blocks of 20 consecutive steps, representing a brief interaction horizon. As shown in Table 16 in Appendix C, step-wise shuffling causes a performance drop, while chunk-wise shuffling matches the unshuffled baseline. This indicates that fully randomizing temporal order is harmful, whereas preserving local structure maintains coordination performance.

**Data Diversity and Scale.** To better understand how CoOT scales with data, we performed ablations on (i) the number of training partners, and (ii) the number of trajectories collected per training partner on Coord. Ring.

As shown in Table 18 in Appendix C, CoOT is more robust to reduced partner diversity, where its performance drops by 24.0% (from 36 to 12), compared to 41.5% for BC. Also, in Table 19 in Appendix C, CoOT degrades more gracefully with fewer episodes (28.6% drop vs. 32.1% for BC).

## 6. Conclusion

We introduced Coordination Transformer (CoOT 🌐), an in-context framework that enables agents to adapt to unseen partners by conditioning on recent interactions. Unlike prior ICL approaches emphasizing task generalization, CoOT targets partner generalization, predicting best-response actions. Experiments on Overcooked and GRF show that CoOT outperforms strong baselines with both evaluation agents and humans, while analyses highlight its rapid adaptation toward non-stationary partners. In summary, this work formulates partner-centric coordination as an in-context learning problem and shows that conditioning on full interaction histories can support effective few-shot adaptation in complex environments. Still, our approach has several limitations. It relies on action histories without explicit communication, and our experiments focus on coordination benchmarks rather than embodied settings with rich perception and control. It also uses a strict no-prior cold-start protocol, so retrieved or pre-filled contexts from similar partners could reduce early interaction costs. Extending CoOT to communication-based and embodied coordination remains an important direction.

## Impact Statement

This work studies in-context partner adaptation for multi-agent coordination, with the goal of making agents more reliable when interacting with unseen teammates. If transferred to real-world settings such as human-AI collaboration, assistive robotics, or multi-robot systems, this capability could help reduce coordination failures and improve safety and efficiency, especially when online fine-tuning is impractical.

We do not anticipate immediate negative societal impacts from the experimental settings considered in this paper. However, similar adaptation mechanisms could be misused in interactive systems. For example, agents may become overly tuned to short-term user behaviors, reinforce unsafe habits, or exploit predictable partners in competitive or mixed-motive environments. More broadly, as with many adaptive learning systems, deployment without supervision may lead to unintended behavior.

We therefore encourage future work to study such methods under safety constraints, with transparent user controls and careful monitoring, especially in applications involving direct interaction with humans.

## Acknowledgments

This work was supported in part by the National Science and Technology Council, Taiwan, under Grants 114-2628-E-002-021- and 115-2634-F-002 -012-, and the Taiwan Centers of Excellence in Artificial Intelligence. Shao-Hua Sun was supported by the Yushan Fellow Program of the Ministry of Education, Taiwan. We thank Ting-Yu Su for the assistance in creating Figure 1.

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

# Appendix

# A. Experiment Environment

## A.1. Overcooked

We evaluated in the Overcooked environment (Carroll et al., 2019) inplemented in ZSC-Eval (Wang et al., 2024)[1]. We used five layouts: Bothway Coordination (*Bothway Coord.*), Coordination Ring (*Coord. Ring*), Counter Circuit (*Counter Circ.*), Asymmetric Advantages (*Asymm. Adv.*), and Coordination Ring with Multi-Recipe (*Coord. Ring multi-recipe*). The multi-recipe layouts have onion (O) and tomatoes (T) as ingredients, which expands the range of recipes from just onion soup (3O) to five types of soups, including mix soup (1O1T), less onion soup (2O), tomato-onion soup (2T1O), onion-tomato soup (2O1T), and onion soup (3O). The following are the details and main challenges for each layout.

**Bothway Coordination.** In this variant, both players can access onions and pots, which broadens the range of feasible strategies and introduces new opportunities for cooperation. This layout helps reduce idle time seen in the Forced Coordination setting and encourages more diverse policies. Still, since plates and the serving station remain confined to one side, effective teamwork is essential to fulfill orders.

**Coordination Ring.** *Coord. Ring* features a compact, circular layout that facilitates close agent interaction. Ingredients, plates, and the serving station are grouped in the bottom-left area, while cooking pots are placed in the top-right. This spatial arrangement drives continuous movement and requires players to coordinate effectively as they manage shared resources and navigate the kitchen.

**Counter Circuit.** Similar in shape to *Coord. Ring*, *Counter Circuit* introduces a central, elongated table, creating narrow pathways that often cause congestion. This layout demands careful movement planning, as agents must avoid blocking one another. A common cooperative tactic involves staging onions in the center to streamline ingredient transfer.

**Asymmetric Advantages.** This layout divides the kitchen into two largely self-contained workspaces while maintaining interdependence through asymmetrically shared resources. Each player has unique access to ingredients and serving stations, while two centrally located pots are jointly accessible. Notably, one player benefits from closer proximity to the serving station, encouraging the development of collaboration strategies to balance workload and improve efficiency.

**Coordination Ring with Multi-Recipe.** This extended version of *Coord. Ring* includes tomatoes positioned near the serving area in the bottom-left corner, increasing task complexity. The added ingredient and recipe variety increase the need for coordinated planning and amplify the importance of cooperation in fulfilling diverse orders. The reward table for this multi-recipe variant is shown in Table 6

## A.2. Google Research Football

**3 vs 1 with Keeper.** In the scenario, three controlled left-team players start near the three edges (left, top, and bottom) of the right-team goal area. The player ($P_0$) standing near the left edge starts with the ball. The two other controlled players ($P_1$ and $P_2$) begin in open passing positions. And a right-team defender, attempting to block their attack, stands between $P_0$ and the goal, along with a right-team goalkeeper. This setup naturally requires multi-agent teamwork among three attackers and introduces longer-horizon coordination compared to Overcooked.

**Behavior-preferring Agents and Best Response.** We define two sets of discrete events for $P_0$ and $P_2$, and generate 144 biased reward functions through linear combinations of these events. Using these rewards, we train $P_0$ and $P_2$ jointly with $P_1$, producing diverse partner behaviors. After training, we collect the trajectories of $P_1$ acting as the best response to each pair of partners. These best-response trajectories serve as the training data for CooT.

**Setup and Metrics.** In our configuration, each episode lasts 200 steps. Agent performance is measured by the goal rate, defined as the number of successful goals achieved within each episode. Our implementation follows the GRF integration in ZSC-Eval. The evaluation partners are constructed using two components: the biased rewards we design (as detailed in Table 7), and the MAPPO training procedure employed in ZSC-Eval's GRF module. We then select 10 behaviorally diverse agents following the same protocol as in the section 5.1. (further details in Appendix B.6).

---

[1] https://github.com/sjtu-marl/ZSC-Eval, with MIT License.

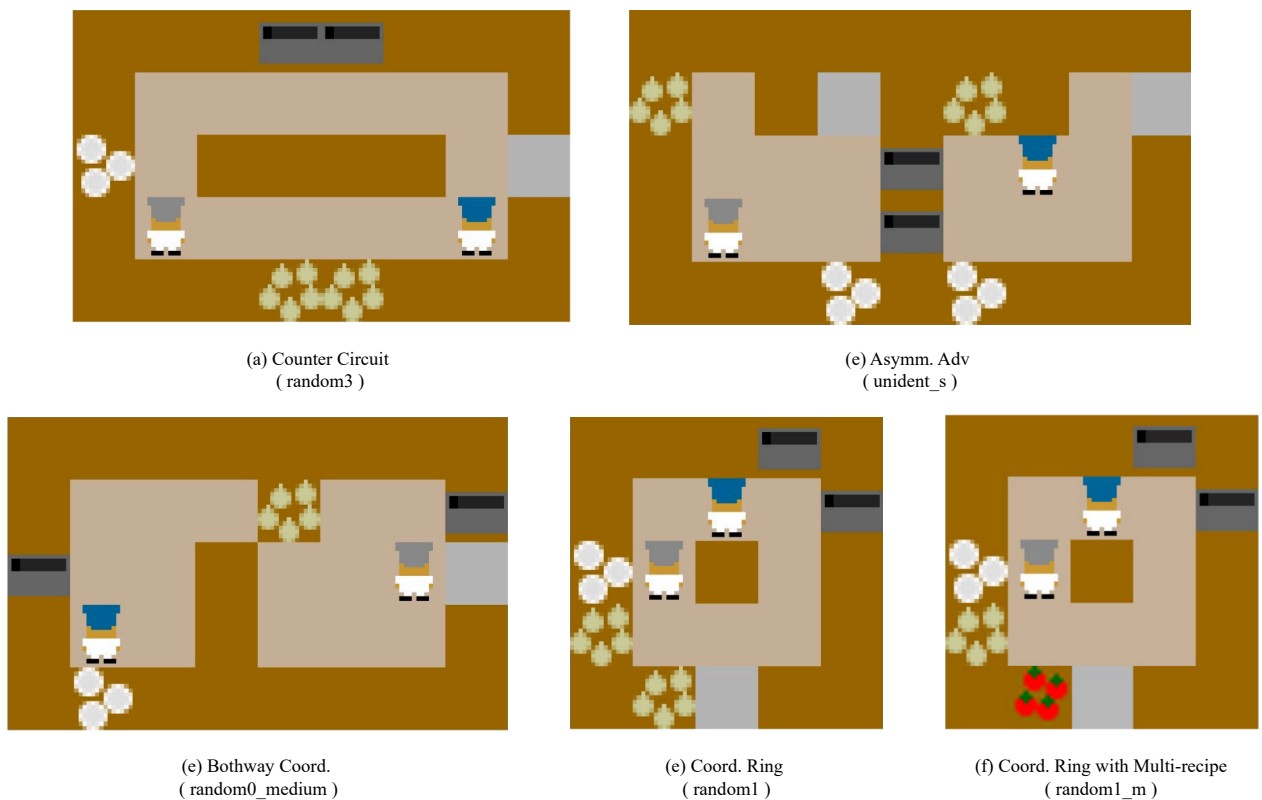

(a) Counter Circuit
( random3 )

(e) Asymm. Adv
( unident_s )

(e) Bothway Coord.
( random0_medium )

(e) Coord. Ring
( random1 )

(f) Coord. Ring with Multi-recipe
( random1_m )

*Figure 3.* **Layouts used in Overcooked.** Each layout introduces unique coordination challenges due to differences in spatial structure and ingredient placement.

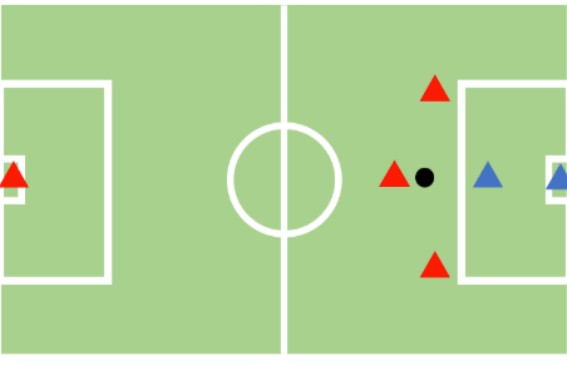

3 vs 1 with Keeper

*Figure 4.* **GRF evaluation scenario.** The *3 vs. 1 with Keeper* scenario requires coordinated offensive behavior, making effective adaptation to partner actions critical for success.

# B. Implementation and Dataset Details

## B.1. Source and Licensing of Policy Pools and Datasets

**HSP and MEP Training Datasets.**   In our setting, we adopt behavior-preferring rewards, which refer to event-based biased reward functions. These rewards assign credit to specific events, such as picking up onions or putting an ingredient into pots. We assume that the behavioral requirements of deployment-time partners can be effectively modeled using this set of reward functions. Details about the reward specifications can be found in Table 5. We defined different sets of event-based rewards according to the characteristics of each layout. For example, the biased agent in Bothway Coordination cannot deliver soup, so we shifted the biased reward from delivering soup to picking up onions or dishes in order to obtain sufficient biased agent policies. In contrast, the biased agent in the multi-recipe *Coord. Ring* can access not only onions but also tomatoes, so some of the events we selected are related to tomatoes.

For two-stage algorithms such as HSP (Yu et al., 2023) and MEP (Zhao et al., 2023), while we do not train these policies ourselves, we construct the candidate policy pool by loading pretrained checkpoints from the ZSC-eval repository, which contains a diverse collection of agents trained under different bias reward settings and algorithmic configurations. These policies exhibit varied behaviors and serve as potential partners for training adaptive agents in the second stage. Following the settings in ZSC-eval (Wang et al., 2024), we selected 36 candidate policies as the basis for partner selection.

*Table 5.* **Events and biased reward under different layouts.** Each entry lists all possible biased reward values that can occur for the corresponding event under the given layout type.

| Events | Bothway Coord. | Multi-recipe | Others |
|---|---|---|---|
| Pickup onions from dispensers | -20,0,10 | 0 | -20,0,10 |
| Pickup dishes from dish dispensers | 0,10 | 0 | -20,0,10 |
| Pickup onion or dish from counters | -20,0 | 0 | 0 |
| Pickup soup from counters | -20,0 | 0 | 0 |
| Put onions into pots | 0 | -20,0 | 0 |
| Put tomatos into pots | | -20,0 | |
| Deliver soup with two ingredients | | -5,0,20 | |
| Deliver soup with three ingredients | | -15,0,10 | |
| Deliver soup | 0 | 0 | -20,0 |
| Stay | -0.1,0,0.1 | -0.1,0,0.1 | -0.1,0,0.1 |
| Order Reward | x0.1,x1 | x1 | x0.1,x1 |

**COOT and BC Training Datasets.**   We construct our expert dataset mainly following the same procedure as in HSP. To build a robust and diverse partner pool $\Pi_{\text{train}}$ for training adaptive policies, we include 36 agents sourced from both MEP and HSP. Specifically, we collect 15 MEP agents with final skill levels, and 21 HSP agents greedily selected based on an event-based diversity score $d_i$, which measures the expected frequency of key events.

For each selected pair $(\pi_i^{\text{p}}, \pi_i^{\text{br}}) \in \Pi_{\text{train}}$, we collect $J$ joint trajectories $\tau$, where $J = 200$ for MEP agents and $J = 250$ for HSP agents (including 220 from final-skill-level and 30 from mid-skill-level checkpoints). While COOT utilizes the current state $s_t$ and the context $C$ to predict $\hat{a}$, the BC baseline is trained using the same dataset by conditioning only on the state $s_t$. Further details of implementation and training will be discussed in the upcoming paragraphs.

## B.2. COOT Training Details

**Dataset Generation.**   To train our Coordination Transformer (COOT), we begin with a set of pretrained policy pairs consisting of biased partner agents and their corresponding best-response policies. For each policy pair, we first select one pair and sample $T$ episodes of rollouts, which we use to construct a context $C$. This process is repeated $K$ times per policy pair to generate $K$ distinct contexts. For each context $C$, we then sample $L$ different query states $s_h$ from the rollout trajectories. Each query state, combined with its associated context $C$ and the corresponding optimal action $\hat{a}$, forms a single training data point of the form $(s_h, C, \hat{a})$. Given a total of $M$ policy pairs, this procedure results in a dataset containing $M \times K \times L$ training examples. More detailed information can be found in Table 10.

*Table 6.* **Reward table for Overcooked *Coord, Ring multi-recipe* layout.** Rewards are assigned based on the ingredient composition of the delivered soup.

| Soup ingredient | Reward |
|---|---|
| onion, onion, onion | $+20$ |
| onion, onion, tomato | $+20$ |
| onion, tomato, tomato | $+20$ |
| onion, onion | $+10$ |
| onion, tomato | $+10$ |
| All other ingredient combinations | $-10$ |

*Table 7.* **Events and biased reward for GRF *3 v.s. 1 with keeper*.** Each entry lists all possible biased reward values that can occur for the corresponding event under the given agent position.

| Events | $P_0$ | $P_2$ |
|---|---|---|
| pass | -1,1 | 0 |
| shot | 0 | -1,1 |
| possession | -0.1,0,0.1 | -0.1,0,0.1 |
| score | 1,5 | 1,5 |

**Online Deployment.** During online deployment, CO0T operates with a dynamically updated context $C$ to enable in-context adaptation. At the start of the first episode, $C$ is initialized as an empty context. Since no prior trajectory is available, CooT selects actions based solely on the current state and the empty context. After completing the first episode, the entire trajectory is stored and used to construct a new context, which is then appended to $C$. As a result, $C$ contains one populated trajectory and $T-1$ empty slots (assuming a fixed context length of $T$). In subsequent episodes, CooT utilizes the current buffer to condition its action predictions, allowing it to adapt based on accumulated experience. The context is managed using a first-in, first-out (FIFO) policy, ensuring that the most recent $T$ trajectories are always retained. This procedure is repeated iteratively throughout the evaluation phase to simulate continual adaptation in a coordination setting. The main method details are provided in Algorithm 1 and Algorithm 2. Algorithm 1 outlines the training process of CO0T, while Algorithm 2 outlines the evaluation process, including how the evaluation partners are selected.

### B.3. Baseline and Training Implementation

Our baseline codebase is primarily based on two open-source frameworks: Imitation[2], an imitation learning library built on Stable Baselines3[3], and ZSC-eval (Wang et al., 2024), a benchmark codebase for Zero-Shot Coordination (ZSC). We use the original implementation of Behavioral Cloning (BC) from Imitation, while the implementations and hyperparameter settings of HSP and MEP are directly inherited from ZSC-eval and HSP (Yu et al., 2023). All of the training hyperparameters are listed in Table 12.

HSP-meta extends HSP by adding a trajectory encoder that encodes the previous interaction trajectory of HSP agents into a latent representation, which is then appended to the agent's observations. This allows the policy to condition on a summary of past partner behavior without modifying the underlying HSP training objective.

To verify that the underperformance of HSP-meta is not due to suboptimal hyperparameters, we train 7 configurations on *Coord. Ring* — one original configuration and 6 additional configurations each varying a single hyperparameter from the original: number of encoder layers, hidden dimension, or learning rate (Figure 5). The best-performing configuration on *Coord. Ring* uses $\text{lr} = 10^{-4}$, which we then evaluate across all five layouts. However, as shown in Table 8, this tuned configuration improves only on *Coord. Ring* and *Asymm. Adv.*, while degrading substantially on the remaining layouts, resulting in lower overall reward and BR-prox than the original. We therefore retain the original hyperparameters in table 12 for all main experiments.

To provide a more comprehensive comparison, we additionally report the performance of HSP and MEP trained with sparse rewards in Table 26.

### B.4. Hardware Specifications and Training Time

We performed the experiments using the workstations listed in Table 9.

Our method takes approximately 19 hours to train for one seed on one layout, assuming a pre-existing set of behavior-preferring agents and their best-response policies. These 19 hours include collecting interaction trajectories from all agent

---

[2]https://github.com/HumanCompatibleAI/imitation, with MIT license
[3]https://github.com/DLR-RM/stable-baselines3, with MIT license

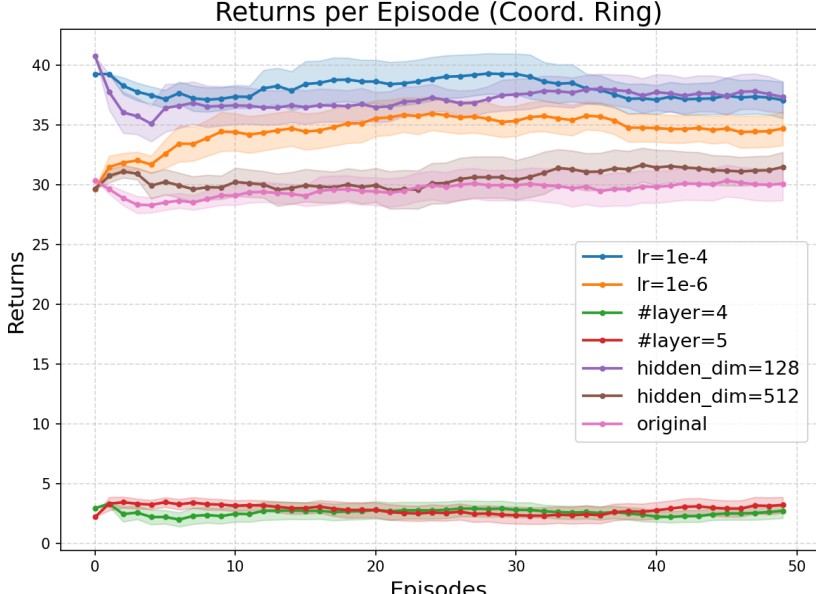

*Figure 5.* **HSP-meta performance under different hyperparameter configurations on *Coord. Ring*.** We train 7 configurations: one original (lr= $10^{-5}$, #layers= 3, #hidden_dim= 256) and 6 variants each varying exactly one hyperparameter (#layers $\in \{2, 4\}$, #hidden_dim $\in \{128, 512\}$, lr $\in \{10^{-4}, 10^{-6}\}$). Each configuration is averaged over 3 random seeds. The best configuration (lr= $10^{-4}$) is further evaluated across all layouts in Table 8.

*Table 8.* **HSP-meta performance under original and tuned hyperparameters across all layouts.** The tuned configuration improves on *Coord. Ring* and *Asymm. Adv.* but degrades on remaining layouts, yielding lower overall performance. We retain the original configuration for all main experiments. **Bold** indicates the better result. **original**: lr= $10^{-5}$, #layers= 3, #hidden_dim= 256. **Tuned**: lr= $10^{-4}$, #layers= 3, #hidden_dim= 256.

| Layout | Coord. Ring | | Coord. Ring Multi-recipe | | Counter Circ. | |
| --- | --- | --- | --- | --- | --- | --- |
| | Return | BR-prox | Return | BR-prox | Return | BR-prox |
| HSP-meta (original) | 29.84±3.92 | 0.35±0.04 | **30.21±1.37** | **0.34±0.02** | **3.28±0.21** | **0.03±0.00** |
| HSP-meta (tuned) | **37.90±4.69** | **0.44±0.05** | 5.74±11.80 | 0.06±0.14 | 2.44±1.32 | 0.02±0.02 |

| Layout | Asymm. Adv. | | Bothway Coord. | | **Overall** | |
| --- | --- | --- | --- | --- | --- | --- |
| | Return | BR-prox | Return | BR-prox | Return | BR-prox |
| HSP-meta (original) | 113.16±12.72 | 0.54±0.06 | **20.44±4.59** | **0.20±0.05** | **40.19±2.28** | **0.29±0.01** |
| HSP-meta (tuned) | **122.40±10.49** | **0.58±0.05** | 19.52±2.88 | 0.20±0.03 | 37.60±4.92 | 0.22±0.04 |

pairs (approximately one hour with 100 parallel environments) and training CooT on the collected dataset. Both stages correspond to single-GPU execution and can be reduced with further parallelization.

Population-based baselines such as MEP and HSP require per-partner adaptation via online cross-training, which takes approximately 5 hours with 100 parallel environments for one seed on one layout. We note that CooT assumes access to a diverse training population, which is a shared assumption across AHT methods, including HSP and MEP.

Reproducing all results, including the baselines, requires approximately 800 GPU hours when running sequentially.

### B.5. Inference Time Comparison

To complement the training cost reported above, we provide a comparison of pure model inference time, excluding environment interaction, across the architectures used in our experiments. Table 11 reports the average per-step inference time. While transformer-based ICL introduces additional overhead compared to lightweight baselines, the runtime remains practical in our multi-agent setting, as CooT requires only 2.41 ms per step, which is acceptable for our simulation-based evaluations.

*Table 9.* **Computational resources used.** Specifications of the four workstations used for training and evaluation, including CPU, GPU, and memory.

| Workstation | CPU | GPU | RAM |
|---|---|---|---|
| Workstation 1 | Intel Xeon W-2255 | NVIDIA GeForce RTX 3080 | 125 GiB |
| Workstation 2 | Intel Xeon W-2255 | NVIDIA GeForce RTX 3090 ×2 | 125 GiB |
| Workstation 3 | Intel Xeon w7-2475X | NVIDIA GeForce RTX 4090 | 125 GiB |
| Workstation 4 | Intel Xeon w7-2475X | NVIDIA GeForce RTX 4090 ×2 | 125 GiB |

*Table 10.* **Dataset configuration for training the Coordination Transformer (CooT).** It specifies the number of partner policy pairs, contexts, rollouts, and resulting total dataset size.

| Dataset | Parameter | Value |
|---|---|---|
| | Number of partner policy pairs ($M$) | 36 |
| | Number of contexts per pair ($K$) | 125 |
| CooT Dataset | Number of rollouts per context ($T$) | 5 |
| | Number of data per context ($L$) | 70 / 200 (GRF) |
| | Total dataset size ($M \times K \times L$) | 315,000 |

## B.6. Evaluation Pipeline Details

To evaluate ad-hoc teamwork with unseen partners, we follow the ZSC-Eval framework (Wang et al., 2024), which constructs a diverse partner population and measures how well an ego agent coordinates with them. Below, we provide the implementation details omitted from the main text.

**Behavior Feature Extraction.** For each candidate partner $\pi_\mathbf{w}$, we compute a high-level behavior feature vector from event occurrences. We define

$$\phi(s_t, a_t) \in \mathbb{R}^m$$

as an *event-based feature vector*, where each component $\phi_j(s_t, a_t)$ indicates whether the $j$-th predefined event occurred at time step $t$ (e.g., pick-up, drop-off, interact-with-object, move-to-location). The overall behavior feature of the approximate best response is

$$\theta_\mathbf{w} = \mathbb{E}\left[\sum_{t=1}^{T} \phi(s_t, a_t)\right],$$

computed over full episodes with partner $\pi_\mathbf{w}$ and its best response.

**Similarity Computation.** Similarity between two partners is defined as the dot product of their behavior features:

$$K_{ij} = \theta_i^\top \theta_j.$$

**Best Response Diversity.** For a subset of partners $\mathcal{S}$, we compute the Best Response Diversity as

$$\mathrm{BR-Div}(\mathcal{S}) = \det(K_\mathcal{S}),$$

where $K_\mathcal{S}$ is the similarity submatrix restricted to $\mathcal{S}$. Larger determinants correspond to subsets whose best responses exhibit more diverse behaviors.

**Partner Selection via Determinantal Point Process.** The partner subsets are sampled using a Determinantal Point Process (DPP) with kernel $K$. Multiple subsets are drawn, and we select the one with the highest BR-Div as the evaluation partner set. Earlier training checkpoints of the selected partners are also included to capture a wider range of skill levels.

*Table 11.* **Average per-step inference time across model architectures.** Reported times reflect pure model forward pass per action, excluding environment interaction.

|  | CooT | MEP | HSP | BC |
|---|---|---|---|---|
| Inference Time (ms) | 2.41 | 1.73 | 1.77 | 0.54 |

*Table 12.* **Training hyperparameters for all methods.** We list model settings for CooT, PPO parameters for HSP/MEP, and additional encoder parameters for HSP-meta.

| Method | Hyperparameter | Value |
|---|---|---|
| BC | Batch size | 256 |
|  | Learning rate | 0.001 |
|  | Optimizer | Adam |
|  | Scheduler | CosineAnnealingLR |
|  | Embedding size | 32 |
|  | Max Epochs | 100 |
|  | Early Stopping Patience | 5 |
|  | Validation Split | 0.1 |
|  | Scheduler $\eta_{min}$ | 1e-5 |
| CooT | Batch size | 120 |
|  | Learning rate | 5e-5 |
|  | Optimizer | Adam |
|  | Scheduler | LambdaLR |
|  | Weight decay | 1e-3 |
|  | Dropout | 0.3 |
|  | Gradient clip norm | 0.25 |
|  | Model | GPT-2 |
|  | Hidden layers | 4 |
|  | Attention heads | 2 |
|  | Embedding size | 128 |
|  | Max Epochs | 70 |
|  | Early Stopping Patience | 25 |
| Stage2 of HSP and MEP | Entropy coefficient | 0.01 |
|  | Gradient clip norm | 10.0 |
|  | GAE lambda | 0.95 |
|  | Discount factor ($\gamma$) | 0.99 |
|  | Value loss | Huber loss |
|  | Huber delta | 10.0 |
|  | Optimizer | Adam |
|  | Optimizer epsilon | 1e-5 |
|  | Learning rate | 5e-4 |
|  | Parallel environment threads | 100 |
|  | Environment steps | 10M |
|  | Reward shaping horizon | 10M |
| HSP-meta | Parallel environment threads | 50 |
|  | Encoder loss | Reconstruction loss |
|  | Encoder layers | 3 |
|  | Hidden dimension | 256 |
|  | Encoder lr | 1e-4 |

**Partner-Specific Best Response Return.**    To provide additional clarity on the pipeline and the main results, we present the **absolute partner-specific best-response return** for each layout in Table 13.

*Table 13.* **Partner-specific Best Response Return.** Note that 10 evaluation partners were used for all Overcooked layouts except *Coord. Ring Multi- recipe.*

|            | Bothway Coord. | Coord. Ring | Coord. Ring Multi-Recipe | Counter Circuit | Asymm. Adv. |
|------------|----------------|-------------|--------------------------|-----------------|-------------|
| Partner 1  | 100 | 80  | 110 | 120 | 220 |
| Partner 2  | 140 | 120 | 90  | 80  | 160 |
| Partner 3  | 100 | 100 | 100 | 60  | 220 |
| Partner 4  | 160 | 100 | 80  | 100 | 220 |
| Partner 5  | 0   | 140 | 90  | 80  | 220 |
| Partner 6  | 100 | 100 | 100 | 60  | 0   |
| Partner 7  | 60  | 100 | 90  | 80  | 220 |
| Partner 8  | 60  | 40  | 80  | 120 | 80  |
| Partner 9  | 140 | 80  | 90  | 80  | 200 |
| Partner 10 | 100 | 0   | 100 | 120 | 200 |
| Partner 11 | –   | –   | 110 | –   | –   |
| Partner 12 | –   | –   | 100 | –   | –   |
| Partner 13 | –   | –   | 90  | –   | –   |
| Partner 14 | –   | –   | 70  | –   | –   |
| Partner 15 | –   | –   | 90  | –   | –   |

## B.7. Pipeline Sanity

**Overlap Between Training and Evaluation Populations.**    To test whether the partner population construction is truly diverse, we conducted (i) a macro analysis, and (ii) a micro analysis, on *Coord. Ring*.

The macro analysis consists of pairing biased agents and their best responders (BRs) across training and evaluation populations, i.e., breaking the original partner-BR pairs.

As Figure 6 shows, only the original pairs (diagonal) achieve high returns, while cross-population returns are low. This indicates that simply memorizing BR behaviors from training partners does not generalize to unseen evaluation partners.

For the micro analysis, we sampled 10k random states and queried actions from the training and evaluation biased agents. We computed the average Jensen–Shannon divergence of **0.3314** between training and evaluation partners, versus 0.2976 between a random policy and evaluation partners. Therefore, we may get the conclusion that training partners are more behaviorally dissimilar to evaluation partners than a random policy would be.

Furthermore, ZSC-Eval (Wang et al., 2024) also provided analyses on high-level behaviors in their Appendix D, confirming such divergence.

**Sensitivity to Evaluation Partner Selection..** To assess sensitivity to partner selection, we re-evaluate CooT and the top-3 baselines on two additional partner sets (Sets 2 and 3), randomly sampled with different seeds instead of DPP selection. As shown in Table 14, relative performance remains stable across sets, indicating that our results are not sensitive to the specific choice of evaluation partners.

## C. Complementary Experiments

### C.1. Context Length

**Experimental Setup.**    We conduct this analysis on Coord. Ring because its overlapping workspaces tightly couple agent actions, making coordination critically dependent on tracking and responding to the partner's evolving behavior. Layouts such as Asymm. Adv. or Bothway Coord. divide agents into separate zones, enabling mostly independent execution with less need for partner-specific adaptation, making them less sensitive to context length choices.

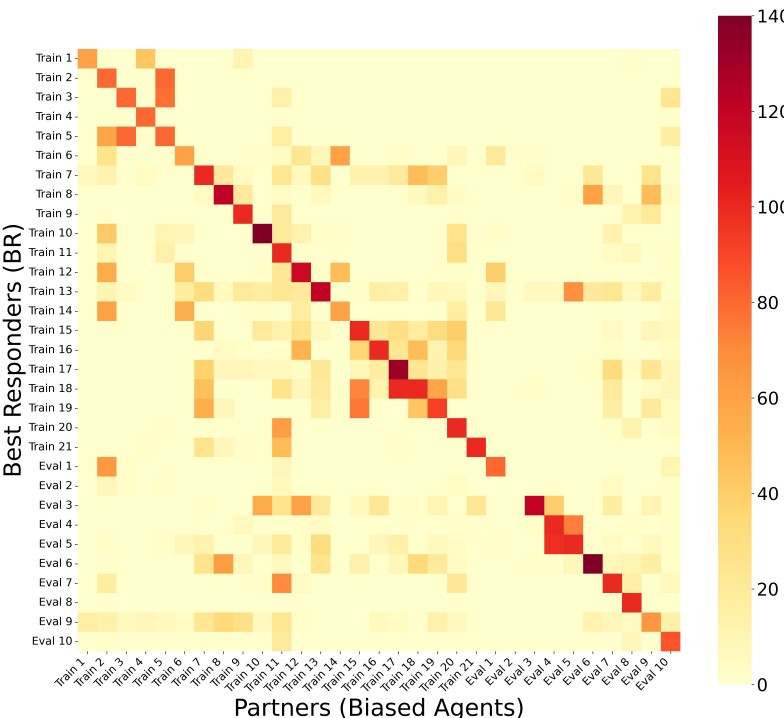

*Figure 6.* **Macro Analysis.** We report the average return over 10 episodes.

*Table 14.* **Sensitivity to evaluation partner selection.** We evaluate CoT and the top-3 baselines on two additional partner sets (Sets 2 & 3) randomly sampled with different seeds, in contrast to the original DPP-based selection (Set 1). Results show stable relative performance across partner sets.

| Method | Set 1 (DPP) | Set 2 | Set 3 |
|--------|-------------|-------|-------|
| BC | $26.24 \pm 1.80$ | $30.83 \pm 1.48$ | $33.78 \pm 2.04$ |
| MEP | $\mathbf{40.30 \pm 3.45}$ | $36.62 \pm 6.23$ | $\mathbf{38.06 \pm 2.10}$ |
| HSP | $\mathbf{41.10 \pm 10.03}$ | $\mathbf{46.47 \pm 1.63}$ | $\mathbf{36.11 \pm 2.55}$ |
| CoT | $\mathbf{38.30 \pm 3.71}$ | $\mathbf{45.96 \pm 4.10}$ | $\mathbf{47.32 \pm 12.85}$ |

**Results.** Performance improves steadily as context length increases up to 5 episodes, reaching the best mean reward of 41.1. Beyond this point, gains diminish and even drop slightly at 7 episodes, indicating that overly long contexts may weaken the influence of relevant information while adding computational overhead. We therefore adopt 5 episodes as the default setting, balancing effectiveness and efficiency.

**Hypothesized Mechanism.** We attribute the degradation primarily to outdated information: since CoT continuously adapts to its partner across episodes, earlier interactions become less informative or even misleading about the partner's current behavior, reducing the model's reliance on more relevant recent context.

**Context Masking as Recency Weighting.** To mitigate the effect of outdated history, we employ context masking during training, which randomly drops a subset of older trajectories, forcing the model to focus on more recent interactions. This constitutes a hard form of recency weighting — older context is removed rather than merely downweighted. Despite this, performance still degrades at 7 episodes, suggesting that masking alone cannot fully compensate for misleading older history. More principled mechanisms — such as attention-based recency weighting or hierarchical context selection — remain promising directions for future work. We adopt 5 episodes as the default setting, balancing coordination effectiveness and computational efficiency.

*Table 15.* **Effect of context length on coordination performance in *Coord. Ring*.** Performance improves steadily as context length increases from 1 to 5 episodes, but gains diminish when extended to 7 episodes.

| Context length | Reward ($\uparrow$) |
|---|---|
| 1 episode | 33.87 |
| 3 episodes | 34.36 |
| 5 episodes | **41.11** |
| 7 episodes | 36.04 |

*Table 16.* **Average episode reward and BR-prox under different shuffle strategies on *Coord. Ring*.** Preserving temporal structure results in better coordination and improved adaptation performance.

| Shuffle strategy | Reward ($\uparrow$) | BR-prox ($\uparrow$) |
|---|---|---|
| No Shuffling | 37.47$\pm$5.67 | 0.44$\pm$0.094 |
| Step-wise | 29.41$\pm$5.23 | 0.36$\pm$0.084 |
| Chunk-wise | **38.30$\pm$3.71** | **0.47$\pm$0.056** |

*Table 17.* **Comparison of online fine-tuning baselines against CooT.** We evaluate MAPPO-based online fine-tuning under different learning rates, comparing *full fine-tuning* (updating all network parameters) and *action-head-only fine-tuning* (updating only the final policy head). Each model interacts with an unseen partner for 50 episodes and is evaluated over 9 partner seeds. Online fine-tuning is unstable: large learning rates often cause collapse, while smaller learning rates do not yield consistent gains.

| Partner | Full fine-tuning | | | | Action-head-only fine-tuning | | |
|---|---|---|---|---|---|---|---|
| | 1e-8 | 1e-7 | 1e-6 | 1e-5 | 1e-8 | 1e-7 | 1e-6 |
| 1 | 7.6 | 6.8 | 6.4 | 1.6 | **8.0** | **8.0** | 6.8 |
| 2 | 6.8 | 7.2 | **8.0** | 5.2 | 6.8 | 6.8 | 6.4 |
| 3 | 38.4 | 39.6 | **41.6** | 32.0 | 39.2 | 39.2 | 38.4 |
| 4 | **38.4** | **38.4** | 37.2 | 32.4 | 36.8 | 36.8 | 36.0 |
| 5 | 26.4 | **30.4** | 28.4 | 16.8 | 28.4 | 28.0 | 28.4 |
| 6 | 40.8 | **44.0** | 38.0 | 31.2 | 41.2 | 41.2 | 40.0 |
| 7 | 37.6 | 35.6 | 32.4 | 32.4 | 37.2 | 38.0 | **38.8** |
| 8 | **47.6** | 46.4 | 43.2 | 31.6 | **47.6** | **47.6** | 46.4 |
| 9 | 20.8 | 20.8 | 19.6 | 15.6 | 21.2 | 20.8 | 20.8 |
| Avg. | 29.4 | **29.9** | 28.3 | 22.1 | 29.6 | 29.6 | 29.1 |

## C.2. Context Shuffling

While CooT demonstrates robust performance in unseen partner coordination, we investigate whether its generalization ability can be further enhanced through improved training-time augmentation strategies. Specifically, we explore whether trajectory augmentation can help the model acquire more robust coordination behaviors. Motivated by Decision Pretrained Transformers (DPT) (Lee et al., 2023), where step-wise trajectory shuffling enhanced generalization in partially observable navigation tasks, we examine whether similar augmentation methods benefit our multi-agent coordination setting. To this end, we evaluate two trajectory-shuffling strategies, step-wise and chunk-wise, in the *Coord. Ring* layout over 50 rollouts each. Although step-wise shuffling has shown benefits in simple navigation environments such as Dark-Room (Zintgraf et al., 2019), where agents operate under short horizons and limited observability, we observe a performance decline when applying this strategy in Overcooked. Unlike Dark-Room's short-horizon navigation tasks, Overcooked requires tight coordination over extended horizons. In such settings, preserving temporal structure remains critical, as effective coordination in Overcooked requires agents to act in a tightly timed relation to one another to avoid unnecessary delays or idle time.

As shown in Table 16, chunk-wise permutation, which shuffles multi-step segments while preserving key temporal dependencies, shows a slight improvement over step-wise shuffling and no augmentation. Interestingly, step-wise shuffling disrupts essential temporal continuity despite increasing data diversity and performs worse than unaugmented data. These findings emphasize the importance of preserving long-range temporal structure in trajectory augmentation, especially for multi-agent tasks that rely on temporally extended interactions.

## C.3. Fine-Tuning

Table 17 illustrates the instability of gradient-based adaptation in coordination settings. While large learning rates (e.g., $10^{-4}$) consistently destabilized training, leading to sharp performance drops, smaller learning rates ($10^{-7}$ and $10^{-8}$) yielded slight improvement. Partial fine-tuning (p-ft) often underperformed full updates, suggesting that restricting adaptation to

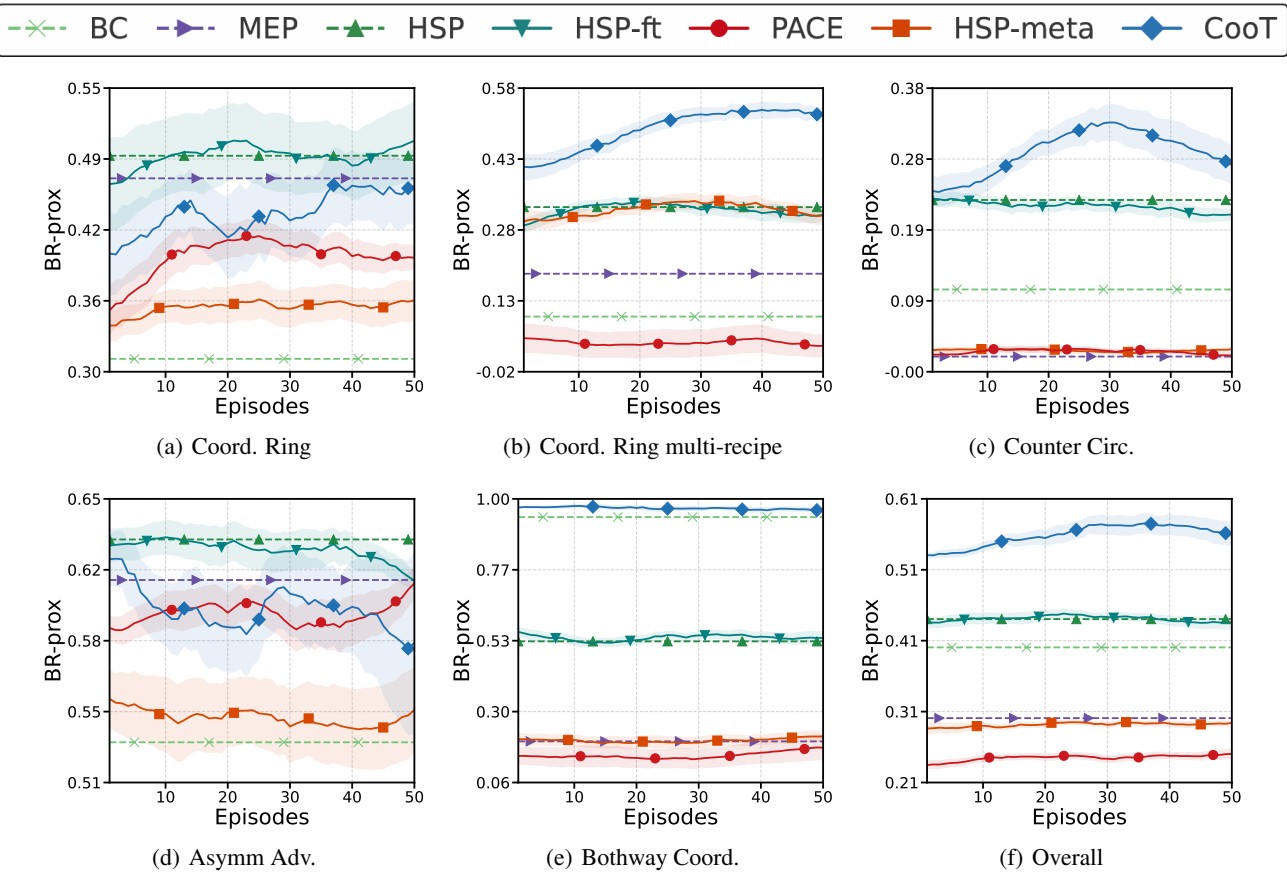

*Figure 7.* **CooT performance across layouts.** Learning curves on six evaluation layouts: Coord. Ring, Coord. Ring Multi-recipe, Counter Circ., Bothway Coord., Asymm Adv., and the aggregate result across all layouts (Overall).

the actor head fails to capture the dynamics needed for partner alignment. Overall, the limited and unstable gains from fine-tuning highlight the contrast with CooT.

### C.4. Adaptation

Figure 7 illustrates adaptation trends as additional interaction episodes are accumulated, comparing CooT against HSP-ft, HSP-Meta, and PACE (with HSP, MEP, and BC included as a non-adaptive reference). Across all five layouts, CooT shows a clear upward trend with more episodes, whereas HSP-ft, HSP-Meta, and PACE remain mostly flat and, in some cases, exhibit unstable trajectories. The gap is most pronounced in *Coord. Ring multi-recipe*: CooT continues to improve throughout the full 50 episodes, but HSP-ft and HSP-Meta saturate early; similarly, PACE does not reliably improve and can fall below the non-adaptive HSP baseline.

These results suggest that the trajectory encoder introduced in HSP-Meta does not provide sufficiently informative latent representations for coordination. One plausible reason is objective mismatch: the reconstruction loss is not aligned with coordination needs, so subtle partner-specific cues may be smoothed out or misrepresented, preventing the policy from reliably differentiating partners. By contrast, CooT conditions directly on observed interactions and learns to map them to best-response actions, leading to consistent adaptation across layouts.

### C.5. Dataset-scale Ablation

We performed ablations on *Coord. Ring* examining two dataset-scale factors: (i) the number of training partners, and (ii) the number of trajectories collected per training partner.

Table 18 demonstrates that CooT exhibits greater robustness to reduced partner diversity, with performance declining by

*Table 18.* **Ablation on scaling down the number of training partners.** Scaled down from 36 to 12, with three training seeds run for each experiment.

| Method | 36 | 24 | 12 |
|---|---|---|---|
| BC | 26.24±1.80 | 21.47±0.31 | 15.34±1.28 |
| CooT (ours) | **38.30±3.71** | **33.66±4.03** | **29.11±1.16** |

*Table 19.* **Ablation on scaling down the number of trajectories per training partner.** Scaled down from 250 to 150, with three training seeds run for each experiment.

| Method | 250 | 200 | 150 |
|---|---|---|---|
| BC | 26.24±1.80 | 23.70±0.96 | 17.81±1.33 |
| CooT (ours) | **38.30±3.71** | **29.72±3.05** | **27.35±1.03** |

24.0% when the number of partners drops from 36 to 12, compared to a steeper 41.5% degradation for BC. Similarly, Table 19 shows that CooT degrades more gracefully under reduced trajectory data, suffering a 28.6% drop versus 32.1% for BC.

## D. Additional Contexts for Human Experiment

### D.1. Experiment Setup

We recruited and verified 36 participants, with a gender distribution of 27 males and 9 females, for the human experiment. Seven participants have prior experience in playing the actual Overcooked!. To mitigate learning effects among the subjects, the order of the agents was randomized. The participants are required to play 200 per episode, 1600 timesteps in total, with 8 episodes for each agent (approximately 5.3 minutes). This leads to a total time of around 30 minutes. The names of the algorithms used by the agents were not visible during the experiments. Agents were differentiated solely by color. Participants were asked to rank the agents after each round, and their trajectories were recorded. All data collection was conducted with the consent of the participants.

### D.2. Experiment Platform

We built our human evaluation platform on top of the ZSC-Eval benchmark (Wang et al., 2024)[A.1], which provides a standardized environment for testing human-AI coordination in Overcooked. To adapt it to our setting, we modified the system to support repeated interactions with the same agent, enabling context accumulation over multiple episodes.

During the experiment, participants controlled one character using keyboard inputs, while the partner was controlled by one of the four agents under evaluation: CooT, MEP, HSP, or BC. To reduce potential bias, agent identities were hidden and replaced with randomized colors. The interface presented real-time feedback, and all trajectories were automatically recorded for later analysis. Figures 8 to 12 show the platform's interface and experiment flow.

### D.3. Additional Human Qualitative Results

To complement the quantitative rankings, we also collected qualitative feedback from participants after each interaction session. These open-ended comments provide insight into subjective impressions of each agent's behavior, including adaptiveness, blocking tendencies, and responsiveness. We organize these comments in Table 22, aligned with the BC, MEP, HSP, CooT order based on participant rankings.

**Qualitative Analysis.** To ensure that CooT's performance is behaviorally grounded rather than a result of a halo effect, we analyze all $N = 144$ open-ended responses. We use Gemini 2.5 Flash to categorize comments along two dimensions: *Explicit Adaptation* (evidence of strategy modification) and *Negative Behaviors* (instances of obstruction or suboptimal coordination). As detailed in Table 20, the model is prompted to establish specific criteria before processing the data. To ensure integrity, a researcher manually audits all model-identified instances to eliminate false positives and reviews excluded responses to prevent false negatives. The resulting distribution (Table 21) aligns with our quantitative rankings, confirming that participants perceive CooT as a more adaptive and smoother collaborator.

*Table 20.* **Prompts for LLM-based qualitative coding.** This table details the specific instructions provided to Gemini 2.5 Flash for categorizing participant feedback. The two-stage process requires the model to first define classification criteria to ensure consistency across the four tested algorithms.

| Category | Prompt Template |
|---|---|
| **Explicit Adaptation** | "Look at the comments for the algorithms (CooT, BC, MEP, HSP), list out the IDs of comments that have explicit mentioning of adaptation, and give a rate of **adaptive** comments for each algorithm. Please first think about your criteria, and then go over all of them one by one." |
| **Negative Behaviors** | "Look at the comments for the algorithms (CooT, BC, MEP, HSP), list out the IDs of negative comments, and give a rate of negative comments for each algorithm. If the comment mentions improvements, ignore it. Please first think about your criteria, and then go over all of them one by one." |

*Table 21.* **Qualitative coding distribution.** Human study comments are first filtered using Gemini 2.5 Flash for adaptation-related and negative content according to predefined criteria, followed by manual verification. $N$ denotes the number of comments.

**(a) Negative behaviors**

| Agent | N | Participant IDs |
|---|---|---|
| BC | 19 | 1, 2, 3, 5, 6, 9, 12, 13, 15, 16, 21, 24, 26, 28, 29, 32, 33, 35, 36 |
| MEP | **11** | 9, 10, 11, 14, 16, 18, 19, 20, 24, 31, 34 |
| HSP | 21 | 2, 3, 4, 6, 8, 10, 14, 15, 17, 20, 22, 23, 24, 26, 27, 31, 32, 33, 34, 35, 36 |
| CooT | **13** | 1, 2, 7, 8, 9, 10, 14, 22, 23, 28, 30, 31, 35 |

**(b) Explicit adaptation**

| Agent | N | Participant IDs |
|---|---|---|
| BC | 3 | 11, 18, 31 |
| MEP | 4 | 6, 21, 30, 32 |
| HSP | 5 | 5, 8, 11, 20, 28 |
| CooT | **9** | 5, 11, 12, 14, 29, 32, 33, 34, 35 |

*Table 22.* **Human feedback comments for different agents.** This table reports representative free-form comments collected from participants during the human–AI evaluation. All comments are anonymized and shown exactly as written, without further editing.

| USER | CooT |
|---|---|
| 1 | It didn't put onion to the pot which already had onion inside. Sometimes it didn't manage to get the onion put on the middle table. |
| 2 | It can understand basic gaming strategy but cannot understand my intention |
| 3 | Do not take onions. More predictable. |
| 4 | better than first |
| 5 | It would block me at first, but it improves collabing with me. I put the onion, and he would take the soup. It would also do my work as well. Generally, it just feels smoother and better along the play. |
| 6 | Disappointed. I found that agent3 is good at the game. However, he refused to adapt to my strategy so waste a lot of time stucking thogather. |
| 7 | agent1 know how to use middle table but often block my way |
| 8 | has some strategies, but some behaviors are meaningless |
| 9 | not very smart |
| 10 | He performs well when playing on his own, so he would take three onions and a plate in the beginning. If I stole his soup, he would freak out and do nothing for a while. However, he walks really fast on his own. |
| 11 | It finds in the begining, I will put onion in the middle table, so it waits for me. Smart! BUT it likes to hold the plate and wait in front of the pot. But this robot is the best to collaborate. |
| 12 | better collaborative, sometime will be at right position |
| 13 | it doesn't put onion into the pot with more onions |
| 14 | A stubbern model. S/he just didn't know how to change the route. However, s/he found my behavior pattern and tried to change his/her behavior. |

| 15 | The agent is smart and sometimes provide hints to me. |
|---|---|
| 16 | often block the road, but seldon hesitate |
| 17 | Agnet4 play well. We play our own rules and get the highest grade. |
| 18 | He keeps take the plate when there is no onion soup ready. |
| 19 | not a speedy start...but I think we'll become a great partner in a near future |
| 20 | so-so |
| 21 | good. |
| 22 | it stop at the first time and it will take the plate at the first time |
| 23 | kinda stupid |
| 24 | Doing well on placing egg, but worse whiling placing dish |
| 25 | hehehe |
| 26 | Pretty good |
| 27 | I think the agent performs best at the beginning. It starts to become confusing after rounds. |
| 28 | Seems not to care much about my play style. Always wanted to achieve its purpose no matter what. Meaning that it would block my way and would not back the fuck off. |
| 29 | The agent can wait me to put onions. |
| 30 | This agent often block my way. We often stood there and look as each other for a long while. It seems that this agent doesn't quite understand maximizing throughput. It usually fill in only one cook and wait there with a plate in hand, leaving the other cook idle. |
| 31 | not helpful |
| 32 | I won't recognize this agent as an AI if I collaborate with it in the real game. As time passed by, the agent tends to be more adative and it won't block my way often. the only point is that he won't put the onion on the oven that already had one or two onion, since it might be more effective and efficient for us to give our meal faster. |
| 33 | At the begining we don't have a good strategy, but we become more efficient and the agent is pretty collaborative |
| 34 | It responded quickly and knowed how to cooperate with me well.. |
| 35 | The agent sometimes block my way. However, it seems to know my playing style and can collabrate with me to some extent. |
| 36 | not bad |

| **USER** | **MEP** |
|---|---|
| 1 | it helps me take the onions to the stove. |
| 2 | It can coorperate with me a little bit in the early episode, but not in the later one |
| 3 | Move with initiative. |
| 4 | quite silimilar to first |
| 5 | The performance is not bad. But it gets a little bit worse over time. The agent performs well, but it can't compromise to my plan. |
| 6 | Agent2 kind of understand my strategy at later runs however the as the score gets higher some unseen situation still confuse him. |
| 7 | agent4 can walk around and not stock traffic |
| 8 | Quick guy, but seems to have some prefer pattern, didn't adapt very much |
| 9 | also not very smart |
| 10 | He seems to be thinking about every step I do, and then think about what he should do. This takes up a lot of time and causes inefficiency. Moreover, I think he loves me and likes to block in my way. |
| 11 | I think this robot play the game itself but this one is smart. |
| 12 | Although it play bad, it improve a lot at the end |
| 13 | i expect it to take a plate when i took the third plate but it didn't |
| 14 | S/he is faster than the former one. However. I tried to collaborate with him/her, but nothing happened. |
| 15 | The agent sometimes blocks me but sometimes is clear and follows my strategy. |
| 16 | doesn't follow the same route, block the way sometimes, |
| 17 | Agent1 is smart sometimes but is also stubid sometimes. |
| 18 | He didn't know that he could take the onion from the middle(which I just put there before). |

| | |
|---|---|
| 19 | non-improving partner; always work in an averaged level |
| 20 | not good at all |
| 21 | can trace me step but sometimes missed |
| 22 | like a human, but it does not learn my work model |
| 23 | not the best, but not that bad |
| 24 | Always block my way to go. |
| 25 | hahaha |
| 26 | Co-work good, but not learning |
| 27 | It would help me take the soup, so for the cooperation, i think it performs good, similar to agent 1. |
| 28 | Blocked my path sometimes, but helped me served onion soup ) |
| 29 | Single mode agent |
| 30 | This aggent seems to know how to collaborate. Behaviors such as filling the cook with onions in it first and taking the onions I put on the table are observed. |
| 31 | keep bump into me |
| 32 | I think this guy is performing better and better and he seems to learn how to use the central block. It performs bad at first, too, but it just become better over time. |
| 33 | At the starting rounds, it seems that we developed a good strategy. However, the agent starts to violate the strategy and frequently results in conflict |
| 34 | Sometimes I was blocked by the agent 1. |
| 35 | It peforms well at the begining, so I feel like it can really collaborate with me. However, it didn't seem to adapt to my playing style and seem to became stupid over time. |
| 36 | better |

| USER | HSP |
|---|---|
| 1 | Sometimes it would help me pick up onions but I think we collaborate well. It even put the soup on the table instead of sending it. |
| 2 | It can understand some parts of the rule but lack of the ability to make right decisions, and it can barely coorperate with me. |
| 3 | spin around for no reason. lack sense of the goal. |
| 4 | useless |
| 5 | Not bad. It's slightly self-centered. It can read my intention in some tasks, but it would also do what it wants in some other scenarios. The adaptation over time is not obvious. |
| 6 | very stubben. I use the same strategy every run but agent1 still do stuffs against me. :( |
| 7 | agent3 can only do deliver well |
| 8 | A little bit dumb, seems to know what I try to do sometime, but slow |
| 9 | becomes worse |
| 10 | He seems to be thinking what he should do and what I've done the same time. This causes difficulties in collaboration because human would decide whether to first observe of first perform action, not doing it in the same time. Eventually, he is blocking my way and doing nothing. |
| 11 | this robot find I prefer to put one onion on the table in the middle first and it did it in the last round. Good. BUT it is stupid than first one. |
| 12 | The performance is set betwwen 1 and 4 |
| 13 | doesn't put onion into the pot with two onions in |
| 14 | S/he seems to block my path several times. |
| 15 | The agent always blocks me and do not follow my action. |
| 16 | folow the same route, but delay a little bit |
| 17 | Agnet2's behavier is wired. In the beginning he know how to collaborate with me putting the onion to the stove, but later he forgot and didn't improve. |
| 18 | He is smart but keeps being in front of me. |
| 19 | clumsy guy; I think he's new in kitchen |
| 20 | not good at first, but imporve after that |
| 21 | faster than me |
| 22 | i think better than the green one, but sometime it will stop and does not work. |

| 23 | such a retard |
|----|---------------|
| 24 | agent seems like walking in circle. |
| 25 | wuwuwu |
| 26 | Pretty bad |
| 27 | It stops moving sometimes and it also blocks my way. i dont think it actually improve over time. |
| 28 | Blocking my path and wouldn't back down, and do not like to serve the onion soup. But over time, it learns to back off when I want to move and serve onion soup. |
| 29 | place onion some where other than oven |
| 30 | This agent seems to be "less confident" on what should do. Hesitations are observed. Path blockage still happens but less frequent than agent 1.. |
| 31 | okay |
| 32 | This AI is quite similar as agent 2, I found it quite confused. Sometimes it respond to my action well sometimes it just put things randomly. Agent 2 often block my way but this guy seems that it is a newbie. Overall the Agent 2 and Agent3 are quite similar. Besides, i have tried to take advantage of the block at the center of the map but both of them can't use it properly. |
| 33 | Sometimes it will block my way and do conflict actions |
| 34 | It seems that agent 2 is a bit srupid that it did'n know how to adjust the order of making dish. Moreover, it often blocked me. |
| 35 | The agent peforms pretty bad, and it didn't adapt to my playing style at all! It often blocked my way and did nothing when I was busy. |
| 36 | bad |

| USER | BC |
|------|-----|
| 1 | It always blocks my way and it would pick up plate when the pot is empty. |
| 2 | It only understand a little bit of the game's rule but being very bad at it |
| 3 | Move with less pattern. |
| 4 | great |
| 5 | The agent is more self centered. It can't really adapt to my behavior and read my intentions. Although there are some improvements along the way, yet it gets worse in the end. |
| 6 | Nice guy. |
| 7 | agent2 often do something weird |
| 8 | has diverse behaviors, but some are meaningless |
| 9 | such an idiot, keep blocking my way |
| 10 | Agent1 performs well when I do new moves like take an onion, but performs bad on deciding actions with the oven. |
| 11 | Although in the beginning, this robot did nothing useful for example, it prefer to put the onion in the different pot and hold the plate in front of an unfill pot, in the last round it figure out I would like to put onion in the middle table therefore, it will wait for onion near the pot. |
| 12 | Not very collaborative |
| 13 | sometimes block my way but not always |
| 14 | S/he didn't block my path and utilized the middle table. |
| 15 | The agent always blocks my strategy and takes the wrong items. It doesn't collaborate with me well. |
| 16 | make the wrong step often |
| 17 | Agent3 can learn how to use the middle area. However, I don't know wht it likes to take the plate when nothing is cooking. |
| 18 | He acts stupid at first but learns very fast. |
| 19 | a very speedy partner; I think I didn't leverage his strategy of putting stuff on the middle island |
| 20 | better than agent1 |
| 21 | bad |
| 22 | sometime the derection for agent1 will be different for me. |
| 23 | the smartest one |
| 24 | always stand on the cooking area. |
| 25 | yayaya |

| | |
|---|---|
| 26 | DOESN'T IMPROVE |
| 27 | It sometimes blocked my way, for the adaptive ability, i think it performs similarly. |
| 28 | Very bad player. At first, would help to change from onion to plate to serve. But over time, it became brain damaged and would block my way and would not do anything. |
| 29 | The agent does not know how to use plates. |
| 30 | It seems that we are working separately. This agent has neither help me nor undo my work.It seems to adapt a little bit but not apparent. |
| 31 | keep learning but not really good |
| 32 | It is more stupid and i can easily distinguidh that it is an AI. Sometimes it did weird behavior, sometimes it just idle and didn't do anything. In the round 7 the AI guy just stop in front of the ai for about 15 mins, while both ovens are filled with soup. I can't do anything then. I didn't see that he improve by any means. It sometimes block my way, more often then agent 1. |
| 33 | Very bad at collaboration. Can hardly develop a playing strategy. Seems to be playing by itself |
| 34 | It responded quickly. |
| 35 | The agents often block my way. It's not really helpful. |
| 36 | Very bad |

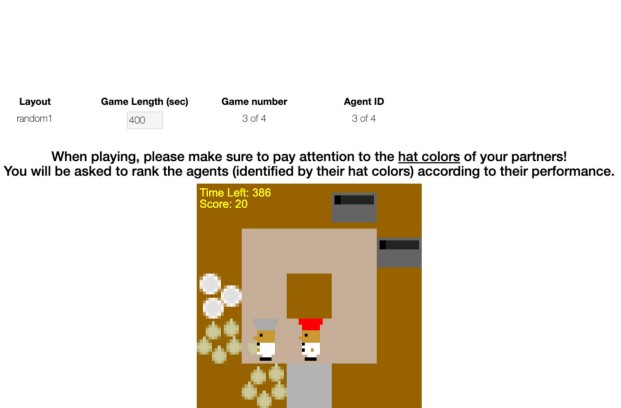

*Figure 8.* **Main experiment layout for human study.** *Coord. Ring* is chosen as it requires both navigation and ingredient coordination, leading to richer coordination scenarios.

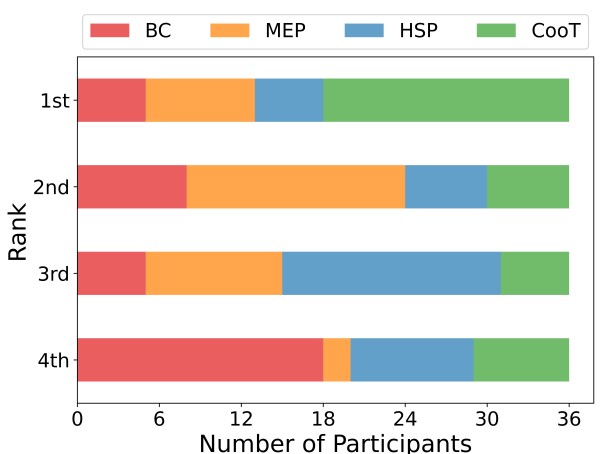

*Figure 9.* **Human study: agent ranking distribution.** Distribution of participant rankings across agents. CooT receives the highest number of first-place rankings.

## Questionnaire

Please rank the agents by dragging the corresponding figures based on your feelings of the agents' cooperation ability.
Please rank the agents **from best to worst, from top to bottom.**

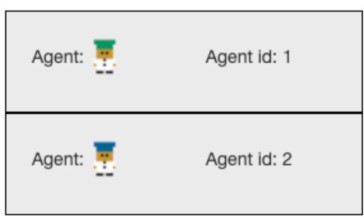

Submit

*Figure 10.* **Human subjective perception ranking system.** Interface used in the user study for collecting participants' subjective rankings of agents. After each round, participants were asked to compare all agents they interacted with and select which partner they preferred most. This ranking procedure complements quantitative metrics by capturing human impressions of collaboration quality.

# Experimental Statement

## 1. Purpose

You have been asked to participate in a research study that studies human-AI coordination. We would like your permission to enroll you as a participant in this research study.

The instruments involved in the experiment are a computer screen and a keyboard. The experimental task consisted of playing the computer game Overcooked and manipulating the keyboard to coordinate with the AI agent to cook and serve dishes.

## 2. Procedure

In this study, you should read the experimental instructions and ensure that you understand the experimental content. The whole experiment process lasts about **30** minutes, and the experiment is divided into the following steps:

(1) Read and sign the experimental statement, and you need to fill in a questionnaire ;

(2) Test the experimental instrument, and adjust the seat height, sitting posture, and the distance between your eyes and the screen. Please ensure that you are in a comfortable sitting position during the experiment ;

(3) You will first try out the game actions you learned in the tutorial within a simple layout to familiarize yourself with the game mechanics;

(4) Start the formal experiment. Please cooperate with the AI agents to get as much scores as possible. You will play with 4 agents in 1 layout. You need to rank the performance of these four agents. After each round, we will ask you to add the current agent to the ranking. After the game ends in each layout, we need to confirm your ranking of the agents.

## 3. Risks and Discomforts

The only potential risk factor for this experiment is trace electron radiation from the computer. Relevant studies have shown that radiation from computers and related peripherals will not cause harm to the human body.

## 4. Compensation

Each participant who completes the experiment will be paid around 6~7 USD.

## 5. Confidentiality

The results of this study may be published in an academic journal/book or used for teaching purposes. However, your name or other identifiers will not be used in any publication or teaching materials without your specific permission. In addition, if photographs, audio tapes or videotapes were taken during the study that would identify you, then you must give special permission for their use.

I confirm that the purpose of the research, the study procedures and the possible risks and discomforts as well as potential benefits that I may experience have been explained to me. All my questions have been satisfactorily answered. I have read this consent form. Clicking the button below indicates my willingness to participate in this study.

*Figure 11.* **Statements for human study.** Consent and instruction form provided to participants before the experiment. It outlines the purpose of the study (human–AI coordination in Overcooked), the procedure (tutorial, gameplay with four agents, and post-round rankings), potential risks and discomforts, compensation details, and confidentiality terms. This ensured that participants were fully informed and agreed to the study protocol before beginning the human–agent collaboration tasks.

## Instructions

### Please read the following instructions carefully.

In this task, you will play in a cooking game as one of the two chefs in a restaurant that serves onion soup. The chef in you control wearing a gray hat.

One of the game layouts looks like:

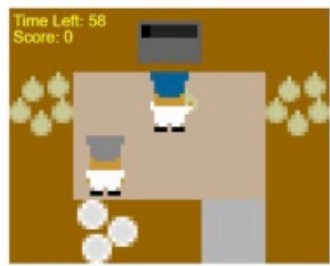

There are a number of objects in the game, labeled here:

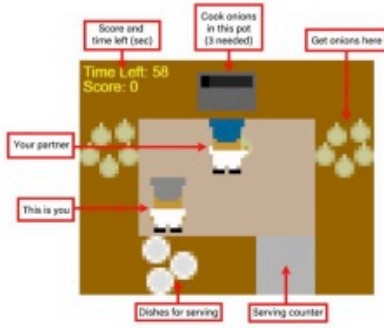

## Movement and interactions

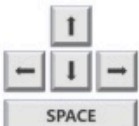

You can move up, down, left, and right using the **arrow keys**, and interact with objects using the **spacebar**.

You can interact with objects by facing them and pressing **space bar**.

Note that you and your partner cannot occupy the same location.

## Cooking

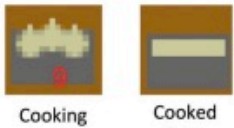

| Cooking | Cooked |
| Soup | Soup |

Once 3 onions are in the pot, the soup begins to cook. After the timer gets to 20, the soup will be ready to be served. To serve the soup, bring a dish over and interact with the pot.

## Goal

Your goal in this task is to serve as many of the orders as you can before each level ends. The current score and time left for you are shown in the upper left of game.

After clicking "Start Playing", you will first play in a warmup trial, where scores will not be recorded.

After the warmup trial, the official experiments will be conducted in 3 layouts. You will complete 7 games with 7 different agents in each layout.

When playing, please make sure to pay attention to the hat colors of your partners! You will be asked to rank the agents (identified by their hat colors) according to their performance.

*Figure 12.* **Instructions for human study.** Guidelines were presented to participants before starting the Overcooked sessions. The instructions described the objective of the task, the number of episodes to be played, and the anonymity of the partner agents (displayed only by color). This ensured participants understood the procedure while minimizing bias toward specific algorithms.

# E. Extended Related Work

**Opponent Modeling.** The development of competitive agents in multi-agent scenarios, especially against unknown and nonstationary opponents, presents a significant challenge. One effective strategy for addressing this is to equip the agent with the ability to model its opponent. This approach, known as opponent modeling, involves conditioning the agent's policy not only on its environment observation but also on predictions about relevant properties of the opponent, such as their policies and goals. Greedy when Sure and Conservative when Uncertain about the Opponents (Fu et al., 2022) solves this by selecting between a real-time greedy policy and a fixed conservative policy using an adversarial bandit algorithm.

**Social Dilemmas.** The concepts of cooperation and competition are fundamental to the study of social systems in both nature and artificial intelligence. Multi-agent reinforcement learning has achieved notable success in settings like Go and Starcraft, which are complex but typically fixed-team and zero-sum. However, the real world often involves mixed-motive interactions that are neither purely zero-sum nor defined by fixed teams. In these settings, agents constantly face social dilemmas, where their individual interests conflict with the collective well-being of the group. Randomized Uncertain Social Preferences (Baker, 2020) solves this by expanding the distribution of environments the agents are trained in, specifically by introducing randomized, uncertain, and asymmetric prosocial preferences. This novel environment augmentation pressures agents to learn socially reactive policies, such as reciprocity and team formation, which are necessary for cooperation.

# F. Aditional Baselines

## F.1. Context-based Meta-RL

We acknowledge that several context-based meta-RL approaches could be applied to the ad-hoc teamwork (AHT) setting. To strengthen our empirical results, we additionally adapt Fast Peer Adaptation with Context-aware Exploration (PACE; Ma et al. 2024) into our cooperative setting. This baseline enables us to more directly compare CooT against a representative context-based approach designed for fast adaptation from recent interaction history.

### F.1.1. METHOD OVERVIEW

**Adapting PACE into HSP.** PACE is a context-aware few-shot coordination method designed to identify and adapt to different peers in multi-agent environments. It uses a context encoder to summarize recent interaction episodes into a latent context. On top of this encoding, a peer classifier predicts the identity or type of the peer and produces an intrinsic exploration reward based on the posterior probabilities of the actual peer agents. The classifier is further trained with an auxiliary loss between the predicted peer distribution and the true peer ID. By conditioning the policy on the peer embedding and using uncertainty-driven intrinsic rewards to encourage informative interactions, PACE enables fast adaptation across diverse peers.

To construct a fair comparison, we integrate PACE's core mechanisms into HSP-meta. Specifically, we add a peer-identifier trained to classify partner policies from the latent context produced by HSP-meta's encoder. The identifier produces a posterior distribution over partner IDs, which we use following PACE's design in two ways: (i) as the target for an auxiliary classification loss, and (ii) to compute an intrinsic exploration reward that encourages the agent to collect interactions that help disambiguate partner identities. During training, this intrinsic reward is combined with the environment reward; at test time, only the latent context (without the exploration reward) conditions the policy. These modifications preserve PACE's identity classification and uncertainty-driven exploration mechanisms while adapting them to the AHT setting.

## F.2. Generative Coordination Methods

Recent work has explored generative approaches for multi-agent coordination, where models synthesize or model diverse partner behaviors for downstream adaptation. To position CooT relative to this emerging direction, we compare against Generative Agent Modeling for Multi-agent Adaptation (GAMMA; Liang et al. 2024) on first-episode and early-episode coordination performance.

### F.2.1. OVERVIEW

**Method.** GAMMA addresses the zero-shot human-AI coordination problem by training a generative model of partner behavior using a Variational Autoencoder (VAE). The VAE encoder infers a latent variable z from coordination trajectories for a partner's unique strategy, skill level, or style. On the other hand, the decoder reconstructs the partner's actions

*Table 23.* **Coord. Ring results for GAMMA.** Evaluated over 10 episodes and 3 random seeds.

| Method | Ep 1 | Ep 1–5 | Ep 1–10 |
|---|---|---|---|
| GAMMA | 29.80±6.07 | - | - |
| CooT | **29.63±4.63** | **39.56±6.46** | **41.11±5.23** |

*Table 24.* **Counter Circ. results for GAMMA.** Evaluated over 10 episodes and 3 random seeds.

| Method | Ep 1 | Ep 1–5 | Ep 1–10 |
|---|---|---|---|
| GAMMA | 9.60±2.42 | - | - |
| CooT | **21.33±6.11** | **21.07±5.69** | **22.73±3.40** |

conditioned on z and interaction history. This generative model can be trained on either simulated agent populations (e.g., from FCP, MEP, or CoMeDi) or human gameplay data, and by sampling from the learned latent space, it produces a richer and more diverse set of training partners than the original data alone. A Cooperator agent is then trained via PPO against these sampled generative partners.

**Implementation.** Following the original setup, we trained GAMMA adaptive agents to converge on each layout and evaluated them using the same partner pool as in our experiments. We follow the official GAMMA implementation and training pipeline, using the recommended VAEs trained on trajectories from a population of 32 agents. The only modification is setting the episode length to 200 (400 in the original paper) to match our environment.

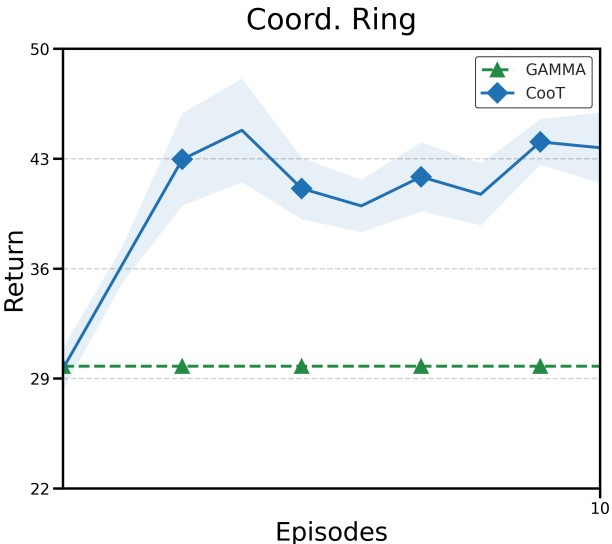

*Figure 13.* **Visualization for GAMMA in Coord. Ring.** Evaluated across 10 episodes, over 3 random seeds.

*Figure 14.* **Visualization for GAMMA in Counter Circ.** Evaluated across 10 episodes, over 3 random seeds.

F.2.2. RESULTS

We report zero-shot performance at episode 1 (Ep 1), averaged over episodes 1-5 (Ep1–5), and averaged over episodes 1-10 (Ep1–10) to capture the few-shot regime in Tables 23 and 24, where all the results are averaged over 3 random seeds (mean ± std). We also visualize the performance of both GAMMA and CooT across 10 episodes in Figures 13 and 14. These results show that CooT matches or exceeds GAMMA in the zero-shot setting (comparable on Coord. Ring and higher on Counter Circ.), while further improving with additional interaction through in-context adaptation. This is consistent with CooT's training: it is exposed to diverse context conditions, including empty-context cases, enabling strong zero-shot performance without sacrificing its ability to adapt online. The difference in evaluation partners is important: the original GAMMA evaluation uses human partners who adapt to the agent, whereas our main evaluation partners have fixed latent preferences and do not react to the agent's behavior. Therefore, it creates a more challenging and diverse coordination setting—especially in Counter Circ., which has a more multimodal strategy space. In this regime, CooT's explicit conditioning on interaction history allows it to infer and adapt to each partner more effectively.

## F.3. Policy-Selection Coordination Methods

We additionally compare COOT against PLASTIC (Barrett, 2015), a representative adaptation method that maintains a Bayesian belief over a fixed library of teammate models and selects among corresponding best-response policies at test time. Unlike PLASTIC, which is fundamentally limited by policies from its training partner set, COOT performs direct in-context adaptation by conditioning on recent interaction trajectories, enabling smoother generalization to previously unseen partner behaviors.

*Table 25.* **Comparison with PLASTIC in Coord. Ring.** We compare COOT against PLASTIC (Barrett, 2015), a Bayesian adaptation baseline that selects among a fixed library of best-response policies at test time. COOT achieves substantially higher return, suggesting benefits from direct in-context adaptation beyond retrieval over precomputed responses.

| Method | Coord. Ring |
|--------|-------------|
| PLASTIC | $19.4 \pm 0.4$ |
| CooT | **38.3 $\pm$ 3.7** |

We implement PLASTIC in *Coord. Ring* and report results in Table 25. COOT substantially outperforms PLASTIC ($38.3 \pm 3.7$ vs. $19.4 \pm 0.4$), suggesting that retrieval over a fixed set of precomputed best responses can be restrictive when evaluation partners exhibit behaviors not well covered by the training library, whereas COOT can adapt its coordination policy online from interaction context.

## F.4. Impact of Reward Shaping

Overcooked environment provides extremely sparse reward, agents can only receive a positive reward upon successfully completing a full sequence of events: place the right ingredients into the pots, pickup plate, and use it to collect the cooked soup from the pots, and deliver the soup. Such delayed feedback requires extensive exploration and often leads to poor performance under limited training steps.

To mitigate this issue, prior work introduces manually designed dense rewards that provide intermediate feedback and guide exploration. However, these dense rewards are unavailable at test time, leading to a mismatch between training and deployment that can significantly degrade performance. To address this, reward shaping is commonly applied by gradually annealing the contribution of dense rewards during training: The shaping coefficient is high in early stages to facilitate exploration and is reduced over time, eventually leaving only the sparse environmental reward. This method encourages efficient learning while preserving test-time compatibility.

To isolate the impact of reward shaping, we additionally design baselines that rely exclusively on sparse environmental rewards throughout training, denoted as **HSP-Sparse** and **MEP-Sparse**. This allows us to evaluate how much of the performance gain in population-based methods can be attributed to dense reward shaping.

### F.4.1. RESULTS

Table 26 compares sparse-only baselines with their dense-reward counterparts. Across most layouts, removing dense reward shaping leads to a substantial drop in performance for both MEP and HSP, highlighting the critical role of dense rewards in facilitating exploration in the extremely sparse Overcooked environment.

When trained exclusively with sparse rewards, population-based methods exhibit limited effectiveness. HSP-Sparse and MEP-Sparse achieve moderate returns only in the simplest layout (Asymm. Adv.), while their performance degrades sharply in layouts requiring tighter coordination, such as Bothway Coord. and multi-recipe settings. This indicates that population diversity alone provides only marginal benefits under sparse feedback.

In contrast, COOT consistently achieves strong performance across all layouts despite being evaluated purely under sparse environmental rewards. Notably, COOT attains substantially higher returns and BR-prox scores in Bothway Coord. and in overall metrics, demonstrating advantages that cannot be explained solely by dense reward shaping. These results suggest that while reward shaping is beneficial for stabilizing training, COOT 's gains primarily arise from its ability to leverage interaction context for adaptation, rather than from auxiliary dense supervision.

*Table 26.* **Additional baselines: HSP-sparse and MEP-sparse.** Performance of population-based methods trained with dense reward shaping, compared across different Overcooked layouts.

| Layout | Coord. Ring | | Coord. Ring Multi-recipe | | Counter Circ. | |
|---|---|---|---|---|---|---|
| | Return | BR-prox | Return | BR-prox | Return | BR-prox |
| MEP-sparse | 4.64±1.41 | 0.05±0.01 | 5.61±10.62 | 0.06±0.12 | 2.36±0.83 | 0.02±0.01 |
| HSP-sparse | 10.21±10.13 | 0.11±0.11 | -1.90±1.65 | -0.02±0.02 | 8.32±4.61 | 0.08±0.05 |
| MEP | **40.30±3.45** | **0.47±0.04** | 16.64±1.16 | 0.19±0.02 | 1.89±0.41 | 0.02±0.00 |
| HSP | **41.10±10.03** | **0.49±0.10** | 29.35±3.77 | 0.33±0.04 | 21.37±1.72 | 0.23±0.03 |
| CooT (Ours) | **38.30±3.71** | **0.47±0.06** | **45.96±3.99** | **0.50±0.04** | **28.28±2.32** | **0.30±0.03** |

| Layout | Asymm. Adv. | | Bothway Coord. | | **Overall** | |
|---|---|---|---|---|---|---|
| | Return | BR-prox | Return | BR-prox | Return | BR-prox |
| MEP-sparse | **133.54±3.81** | **0.66±0.03** | 13.07±3.66 | 0.11±0.04 | 33.97±3.25 | 0.19±0.02 |
| HSP-sparse | 130.90±5.18 | 0.63±0.02 | 41.84±37.04 | 0.40±0.33 | 38.77±7.41 | 0.25±0.07 |
| MEP | 127.44±5.66 | 0.61±0.03 | 22.76±5.59 | 0.20±0.05 | 41.81±0.79 | 0.30±0.00 |
| HSP | **134.01±2.19** | **0.63±0.02** | 54.99±3.56 | 0.53±0.03 | 56.16±2.38 | 0.44±0.02 |
| CooT (Ours) | 129.48±9.34 | 0.62±0.05 | **101.93±1.00** | **0.96±0.01** | **68.79±2.33** | **0.57±0.02** |

# G. Relationship Between Behavior Cloning and CooT

A natural question is whether CooT is simply behavior cloning with a Transformer backbone. While both BC and CooT are trained with supervised objective on best-response demonstrations, their learning formulations differ fundamentally.

**Behavior Cloning.** BC learns a direct mapping from state to action, $s_t \rightarrow a_t$, or, in recurrent variants (BC-RNN), from within-episode history to action, $(h_t, s_t) \rightarrow a_t$, where $h_t$ summarizes past observations and actions sequentially. In both cases, BC learns a *partner-agnostic* policy: the same state produces the same action regardless of who the partner is, since no partner-specific information is available at test time.

**CooT.** In contrast, CooT learns $(C, s_t) \rightarrow a_t$, where $C$ is a *cross-episode* interaction context collected from previous episodes with the same partner. Importantly, during training, query states $s_t$ are sampled independently from the full offline dataset ($s_t \sim \mathcal{D}$), rather than from the temporal continuation of $C$. This prevents the model from solving the task via simple next-step prediction or trajectory completion, forcing it to instead use $\mathcal{C}$ to infer the partner's latent behavioral preferences.

This distinction has two practical consequences. First, CooT produces *partner-specific* responses: the same state $s_t$ can elicit different actions depending on the observed partner context, whereas BC always predict the same action distribution.

Second, the two methods differ fundamentally in how prediction errors accumulate. BC learns a fixed mapping along sequential trajectories, so errors at one timestep shift the agent to states outside the training distribution, causing mistakes to compound over time. CooT is more robust to this for two reasons. First, because query states $s_t$ are sampled *independently* from the dataset during training — decoupled from the context trajectories — the model learns to jointly condition on both the current state and context, rather than relying on sequential state continuations. This means that even when a previous action was suboptimal, CooT can still leverage the interaction context to infer partner behavior and select a reasonable action, preventing errors from propagating. Second, and more importantly, CooT's cross-episode context conditioning enables posterior sampling across episodes: each new episode adds interactions to the context, providing more information about the partner's behavior. Formally, the information gain grows as $O(K)$ in the number of context episodes $K$, while the cumulative cost of imperfect predictions grows only as $\tilde{O}(\sqrt{K})$ (Corollary 6.2 in Lee et al. 2023). Since information accumulates faster than errors do, the average per-episode regret decreases as $\tilde{O}(1/\sqrt{K})$, meaning CooT converges toward optimal behavior as more episodes are observed. This is the *opposite* of compounding error, where mistakes accumulate faster than the agent can correct them, and is directly supported by Figure 2, which shows CooT's coordination performance steadily improving over episodes.

We note that intra-episode compounding errors may still occur within a single episode, which we acknowledge as a residual limitation shared with other imitation learning approaches (Schaal, 1996; Lee et al., 2021; Shafiullah et al., 2022; Chen et al., 2024; Lai et al., 2024; Huang et al., 2024; Yeh et al., 2025).

# H. Algorithm

---

**Algorithm 1** Agent Pool Construction and Training

---

1: // Generating agent pool
2: Initialize empty pool $\Pi_0$
3: **for** $i$ in $P$ **do**
4:     Sample hidden reward function $r_i^w$ from reward space $R$
5:     Train $\pi_i^{\mathrm{p}}$ and its best response $\pi_i^{\mathrm{br}}$ using PPO
6:     Add $(\pi_i^{\mathrm{p}}, \pi_i^{\mathrm{br}})$ to $\Pi_0$
7: **end for**
8: // Partner selection for training
9: Initialize empty training pool $\Pi_{train}$
10: **for** $(\pi_i^{\mathrm{p}}, \pi_i^{\mathrm{br}})$ in $\Pi_0$ **do**
11:     Rollout trajectories and Compute event-based diversity $d_i$ of $\pi_i^{\mathrm{br}}$
12: **end for**
13: Select top-M agents with highest $d_i$ values as $\mathcal{S}$
14: Add corresponding agents to $\Pi_{train}$
15: Remove corresponding agenst from $\Pi_0$
16: // Construct training dataset
17: Initialize empty dataset $D$
18: **for** $(\pi_j^{\mathrm{p}}, \pi_j^{\mathrm{br}})$ in $\Pi_{train}$ **do**
19:     **for** $k$ in $K$ **do**
20:         Rollout $T$ trajectories as context $C$
21:         **for** $l$ in $L$ **do**
22:             Sample query state $s_h \sim C$
23:             Let $a^\star = \pi_j^{\mathrm{br}}(s_h)$
24:             Add data $(s_h, C, a^\star)$ to $D$
25:         **end for**
26:     **end for**
27: **end for**
28: // Model training
29: Initialize model $M_\theta$
30: **while** not converged **do**
31:     Sample $(s_h, \tau_j, a^\star)$ from $D$
32:     Predict action distribution $\hat{p} = M_\theta(\cdot \mid s_h, \tau_j)$
33:     Compute loss $\mathcal{L}$ given $\hat{p}$ and Update $\theta$
34: **end while**

---

**Algorithm 2** Evaluation and Online Deployment (Wang et al., 2024)

---

1: // Evaluation partner selection
2: **for** $(\pi_i^{\mathrm{p}}, \pi_i^{\mathrm{br}})$ in $\Pi_0$ **do**
3:     Rollout trajectories $\tau_i$
4:     Embed features of $\tau_i$ into $\phi_i$
5: **end for**
6: Compute similarity matrix $\mathbf{K}$ from $\{\phi_i\}_{i=1}^K$
7: Sample subset $\mathcal{S}$ from top-N candidates of $\mathbf{K}$
8: Define evaluation set $\Pi_{\mathrm{eval}} = \{\pi_s^{\mathrm{p}}\}_{s \in \mathcal{S}}$
9: // Online deployment
10: Sample unseen partner $\pi_s^{\mathrm{p}} \sim \Pi_{\mathrm{eval}}$
11: Initialize fixed-length context $C = \{\}$
12: **for** episode = 1 to $E$ **do**
13:     **for** timestep $t = 1$ to $Z$ **do**
14:         Observe $s_t$, predict $a_t \sim M_\theta(\cdot \mid s_t, C)$
15:         Execute $a_t$ with partner, observe $(s_t, a_t, r_t)$
16:     **end for**
17:     Append episode trajectory to context $C$
18: **end for**

---

## I. The Use of Large Language Models

We used large language models (LLMs) in limited ways that did not affect the scientific contributions of this work. Specifically, LLMs were employed to (1) polish and improve the clarity of writing without altering the technical content, (2) help organize and summarize qualitative feedback collected from human study participants, and (3) assist in designing and refining figures for presentation purposes. All conceptual, methodological, and analytical contributions, including study design, data analysis, and interpretation of results, were carried out solely by the authors.

