# OpenReview forum: "CooT: Learning to Coordinate In-Context with Coordination Transformers"
_ICML.cc/2026/Conference — ICML 2026 regular_

### Official Review · Reviewer_N4xB · 2026-03-12

**Soundness:** 3
**Presentation:** 3
**Significance:** 3
**Originality:** 3
**Overall Recommendation:** 4
**Confidence:** 4

**Summary:**

This paper proposes Coordination Transformers (CooT), a framework for multi-agent coordination with unseen partners using in-context learning (ICL). They train a GPT-2-based transformer on trajectories from behavior-preferring agents paired with their best-response counterparts, then at test time adapt to new partners purely by conditioning on accumulated interaction history, without any parameter updates. The problem is formalized as a Hidden-Utility Markov Game (HU-MG). The method is evaluated on Overcooked (5 layouts) and Google Research Football (3v1 with keeper), compared against behavioral cloning and other population based MARL approaches. A human study with 36 participants is also conducted.

**Compliance With Llm Reviewing Policy:**

Affirmed.

**Key Questions For Authors:**

- How does CooT perform when tested against partner types not represented in the training distribution — e.g., human-like heuristic agents or scripted agents with qualitatively different behavior profiles? The current evaluation partners are all PPO-trained agents with event-based biases.
- The paper states that context length beyond 5 episodes degrades performance (Table 12). What is the hypothesized mechanism — is this due to stale information, computational saturation, or training distribution mismatch? Have the authors experimented with attention masking or recency weighting within the context window?
- I'd be curious to understand why the authors do not consider more complex ZSC testbeds such as Habitat 3.0: https://arxiv.org/abs/2310.13724 (also see earlier work on ZSC using embodied environments: https://arxiv.org/abs/2306.00087).
- In Asymm. Adv., HSP achieves higher mean return (134.01 vs. 129.48) and comparable BR-prox. In Coord. Ring, HSP matches or slightly exceeds CooT within standard deviation. The paper bolds results within one standard deviation of the best, which is reasonable, but the narrative in Section 5.2.1 may be overclaiming (CooT "consistently outperforms" baselines). It appears that the advantage is concentrated in coordination-heavy settings, shouldnt this be the primary claim?
- The paper notes that HSP-meta and PACE underperform vanilla HSP. Is this attributable to the encoder architecture, the auxiliary losses, or training instability? Clarifying this would help the community understand why compressed context representations fail here.
- The diversity score d_i used for partner selection is described only briefly; a formal definition in the main text (not just the appendix) would improve clarity. Also, how sensitive are results to the diversity selection procedure (DPP-based partner selection for evaluation)?

**Limitations:**

The paper does not discuss strategies to mitigate context cold start (e.g., prior context from similar domains, meta-initialization). Discussion of computational cost asymmetry relative to baselines would also be helpful particularly for real-world deployments. Also, other impacts to consider:
- Positive potential in assistive robotics, multi-robot systems, and human-AI collaboration
- A concrete misuse concern: adaptive systems could become "overly tuned to short-term user behaviors, reinforcing unsafe habits"
- The risk of exploiting predictable partners in competitive or mixed-motive environments.

**Strengths And Weaknesses:**

### Strengths:
- Well-motivated and articulated problem framework: In-context learning for partner generalization instead of task generalization is well explained and supported.
- Good clarity of exposition: The paper is well-written, the diagrams and website are useful to understand the core pieces of the work quickly.
- Ablations on context length and chunk-wise vs. step-wise shuffling and human evaluation are useful in understanding the strength of the approach.

### Weakness:
- Limited novelty: The core architecture: a transformer trained via behavior cloning on (state, context, best-response action) tuples is a relatively direct application of Decision-Pretrained Transformers (Lee et al., 2023) and Algorithm Distillation (Laskin et al., 2023) to multi-agent coordination. The HU-MG formalism is directly borrowed from HSP (Yu et al., 2023). The paper would be strengthened by being crisp about the methodological contribution. While I don't necessarily think methodological novelty is a necessity for acceptance, being explicit/clear about this would be helpful.
-  Potentially unfair baseline comparisons: CooT has a structural advantage over population-based baselines: it receives a rolling context window of recent trajectories at test time, while MEP and HSP operate without any episodic memory. This is a meaningful asymmetry. The paper does not adequately discuss how much of CooT's gain is attributable to the in-context architecture versus simply having access to runtime interaction history. A stronger baseline would be a simpler method (e.g., recurrent policy or a mean-pooled trajectory encoder) that also has access to episodic context.
- Training/evaluation population overlap is unclear: The training pool Π_train is drawn from the same ZSC-Eval repository as the evaluation partners. While evaluation partners are selected via DPP from Π_0 \ Π_train, the exact degree of behavioral overlap between training and evaluation partners is not quantified. It is unclear how much CooT's performance reflects genuine generalization versus interpolation within a familiar behavior space. Experiments with a held-out partner distribution (e.g., human-trained or LLM-driven agents) beyond the human study would strengthen this claim.
- No ablation on the transformer architecture: The model uses a GPT-2 variant with 4 layers, 2 attention heads, and 128-dimensional embeddings. There is no ablation varying model capacity, nor comparison to non-transformer context aggregation methods (e.g., LSTM-based encoders). It is unclear whether the ICL behavior is genuinely emergent from the transformer structure or whether simpler architectures would perform similarly.

---

> ### Author Rebuttal · Authors · 2026-03-31
>
> We sincerely thank the reviewer for the thorough and constructive comments.
>
> > Generalize to OOD partners, e.g., human-like heuristic agents
>
> The human study in Section 5.3 shows that CooT achieves the best performance and is most preferred by the users with non-stationary behaviors extending beyond RL agents.
>
> > Degradation beyond 5 episodes? Attention masking or recency weighting?
>
> We use context masking: we randomly drop a subset of older trajectories during training, forcing models to focus on recent ones. This is a hard form of recency weighting, in contrast to the soft version, which downweights (rather than removes) older context. Despite this, performance degrades at larger context lengths. As the reviewer notes, more principled mechanisms (e.g., attention masking or hierarchical context selection) are promising future directions.
>
> >  Complex embodied 3D testbeds
>
> While embodied testbeds are interesting, our work focuses on how agents adapt to unseen partner behaviors rather than on challenges posed by high-dimensional perception and continuous control in embodied testbeds.
>
> > Primary claim suggestion for sec 5.2.1
>
> We will revise Section 5.2.1 to avoid overclaiming as suggested by the reviewer.
>
> > Why does HSP-meta/PACE underperform vanilla HSP?
>
> We hypothesize
> - The reconstruction loss for learning the context encoder favors all trajectory features equally, failing to capture representative behaviors.
> - During testing, the context encoder may not generalize to unseen partners, produce misleading context, and misguide the policy.
>
> > Evaluation partner selection sensitivity
>
> We additionally evaluated CooT and the top-3 baselines on 2 randomly sampled partner sets (Sets 2 & 3) in Coord. Ring (in contrast to the DPP selection used originally). The results below show stable relative performance across all sets.
>
> |Method|Set 1 (original)|Set 2|Set 3|
> |:-:|:-:|:-:|:-:|
> |BC|26.24±1.80|30.83±1.48|33.78±2.04|
> |MEP|**40.30±3.45**|36.62±6.23|**38.06±2.10**|
> |HSP|**41.10±10.03**|**46.47±1.63**|**36.11±2.55**|
> |CooT (ours)|**38.30±3.71**|**45.96±4.10**|**47.32±12.85**|
>
> > Limitations: compute cost, cold-start, & broader impacts
>
> - **Compute cost**: Table 11 (Appendix) shows that CooT has slightly higher inference time (2.41 ms vs. 0.54-1.77 ms).
> - **Cold start**: We evaluated CooT under a no-prior setting, i.e., adaptation relies purely on online interaction. CooT is compatible with initialization strategies, such as seeding the context buffer with retrieved trajectories from similar partners, which could potentially alleviate the cold-start problem.
> - **Broader impacts**: We will revise the paper to discuss both positive applications (e.g., assistive robotics, human-AI collaboration) and potential risks (e.g., mixed-motive settings).
>
> > Clarifying novelty and contributions
>
> We agree that our contribution is not an architectural novelty. Our contribution is the formulation: we cast ad-hoc teamwork as encoding partner behaviors from interactions and predicting the best response using a Transformer, enabling test-time gradient-free adaptation, whereas prior work mostly uses Transformers for single-agent RL or policy distillation.
>
> > Unfair comparisons: MEP and HSP operate without any episodic memory
>
> The comparison is fair, as HSP and MEP are RNN policies conditioning on past interactions.
>
> > Training/evaluation population overlap
>
> We conducted analyses to quantify discrepancies between the training and evaluation populations on Coord. Ring.
>
> **Macro analysis (return).** We evaluated by pairing biased agents and their best responders (BRs) across training and evaluation populations, i.e., breaking the original partner-BR pairs. We reported the average return over 10 episodes. [The results](https://shorturl.at/lQsJb) show that only the original pairs (diagonal) achieve high returns, while cross-population returns are low. This indicates that simply memorizing BR behaviors from training partners does not generalize to unseen evaluation partners.
>
> **Micro analysis (action divergence).** We sampled 10k random states and queried actions from the training and evaluation biased agents. We computed the average JS divergence: 0.3314 between training and evaluation partners, versus 0.2976 between a random policy and evaluation partners, i.e., training partners are more behaviorally dissimilar to evaluation partners than a random policy would be.
>
> Both analyses show that the training and evaluation populations are behaviorally distinct. We thank the reviewer for the insightful suggestion and will include this in the revised paper.
>
> > Ablation on model capacity and non-Transformer methods
>
> Prior work [1] shows that Transformers are more effective in-context learners than alternatives such as LSTMs. Thus, we adopt a Transformer as our backbone following the architecture of the decision pre-trained transformer.
>
> **References**
>
> [1] Miller et al. "Counterfactual reasoning: an analysis of in-context emergence." NeurIPS, 2025.

---

> > ### Author Rebuttal · Reviewer_N4xB · 2026-04-05
> >
> > Thank you for addressing my concerns. I still remain positive about the paper particularly if the authors can make the proposed changes, add these additional experiments/results, adjust the claims, and clarify novelty focus.

---

> > > ### Author Response · Authors · 2026-04-07
> > >
> > > We sincerely thank the reviewers for the thoughtful review and for engaging promptly with our rebuttal with positive acknowledgment—we really appreciate the time and care you put into our work.
> > >
> > > To further strengthen the paper in response to your feedback, we have incorporated additional experiments and clarifications/ discussions to address the key concerns you raised.
> > >
> > > We added new experiments on partner selection sensitivity, showing consistent relative performance across multiple randomly sampled partner sets beyond the original DPP-based evaluation. We also conducted new analyses to quantify the gap between training and evaluation populations, including both cross-pair return evaluation and action-level JS divergence, demonstrating that evaluation partners are behaviorally distinct and that performance cannot be explained by memorization or interpolation.
> > >
> > > We also provided clarifications/discussions on OOD generalization (via the human study), the mechanism behind context length degradation (including our use of context masking), the scope of evaluation, appropriate calibration of claims, the behavior of HSP variants, the fairness of comparisons, the positioning of our contribution, and the discussion of limitations and broader impacts.
> > >
> > > We thank the reviewer for helping us to clarify, reflect, and improve our work. We will incorporate all the discussion in the revised paper.
> > >
> > > Given that your acknowledgment indicates all the concerns have been addressed, “*(a) Fully resolved - My concerns have been adequately addressed*,” we would be very grateful if you could consider updating the score accordingly.

---

### Official Review · Reviewer_Cbxd · 2026-03-12

**Soundness:** 3
**Presentation:** 4
**Significance:** 2
**Originality:** 2
**Overall Recommendation:** 4
**Confidence:** 5

**Summary:**

The paper addresses the ad hoc teamwork (AHT) problem; specifically, the problem of designing an ego agent that can perform online adaptation to unseen teammates. The paper proposes Coordination Transformers (CooT), which consists of training a transformer policy on an offline dataset of diverse teammates and their best responses, using a supervised, maximum likelihood objective. The key idea is to leverage the in-context adaptation ability of transformers to coordinate with unseen teammates. CooT is evaluated on Overcooked and Google Research Football, as well as with human collaborators. In most scenarios, CooT demonstrates improvements over baseline methods, and is the most preferred by human collaborators.

**Compliance With Llm Reviewing Policy:**

Affirmed.

**Final Justification:**

The authors' rebuttal addressed my main concerns and clarified an important misunderstandings I had around the method. As there are very few papers that demonstrate adaptation between episodes within ad hoc teamwork, the paper's results are likely to be interesting to the AHT community.  Given this, I have decided to increase the score of the paper.

**Key Questions For Authors:**

See above.

**Limitations:**

The authors do acknowledge a limitation of the method (does not address communication), but it is orthogonal to the main idea of the paper. I think there are major limitations that remain unacknowledged. For example, one major limitation is the requirement of a dataset generated by diverse partners and their respective best responses. Additionally, as a method based on BC, naturally CooT should suffer compounding error, especially as the trajectories grow longer / more agents are involved.

**Strengths And Weaknesses:**

**Strengths**

- **Well-written**: the paper is clear and easy to understand.

- **Evaluation on comprehensive benchmark tasks**: the method is evaluated on a number of standard benchmark AHT tasks and includes a human evaluation.

- **Addresses an understudied problem**: I agree that most existing work in AHT ignores the problem of designing an agent capable of online adaptation to unseen teammates, and that this is an important step for AHT.

- **CooT naturally handles partner switching**: in my opinion, it is a strength that CooT can handle partner switching without specialized objectives. There are some prior methods that have been specially designed to solve this subproblem (e.g. [1]), so it’s very interesting that this simple method naturally has this property. I am curious whether CooT can handle partner switching within episodes [1, 2].


**Weaknesses**

- **Methodological Novelty/positioning**

    - Although the study is largely well-done, and the problem has not been studied much, the technique is not very novel. As the authors discuss, the in-context learning ability of transformers has been thoroughly studied. Further, the method appears to amount to doing BC with a transformer policy, which I don’t think can be called a novel method.

    - Further, transformers have been studied for decision-making in  single-agent RL, MARL [3], and AHT [4], where in particular, [3] demonstrated that transformers can represent and deploy various best response policies in a multi-agent scenario.

    - In light of the above related work, I think this paper is better positioned as a *study* of the capabilities of transformers in AHT settings, perhaps similar to [5]

- **Scalability**:

    - The proposed method for test-time adaptation consists of conditioning the CooT on a context consisting of the previous T joint trajectories. As the number of agents scales, the context length will grow linearly. Perhaps due to this, most of the evaluation is on 2-player games.

    - Section 5.2.2 claims to demonstrate the scalability of CooT via evaluation on GRF, but the GRF task is only 3 agents so I don’t feel that it sufficiently demonstrates scalability.

- **Key missing baselines/unfair comparisons:**

    - The paper does a good job of comparing against a variety of alternative strategies  for approaching this problem (BC, population-based methods, meta-learning, gradient-based online fine-tuning) but I still think there are some important yet simple baselines missing.

    - In particular, it’s absolutely essential to compare this approach against recurrent BC (i.e. BC implemented with an RNN, where the RNN should be trained on the same dataset format as CooT). Since this isn’t discussed in the paper, I’m guessing that the existing BC baseline is not recurrent, which is unfair.

    - CooT relies on a manually curated set of diverse teammates to generate data to train with. So, there should be a baseline where CooT is compared to an RL policy trained directly against that set of curated diverse teammates for fairness. The existing MEP/HSP baselines are not equivalent to this baseline because they generate their own teammates -- therefore, the failure of MEP / HSP could be a failure to generate diverse teammates.

    - Additionally, there should be a comparison against a classic, playbook style method called PLASTIC-Policy [6, Alg. 4] that is similar to CooT. The main idea in PLASTIC is to learn a set of best responses to a set of diverse training teammates and a Bayesian best-response selector that selects the appropriate best response at test time. [6] demonstrates that PLASTIC also improves against unseen teammates at test time. In my view, CooT and PLASTIC are in the same family of AHT methods,  where the main difference is that CooT uses a single transformer policy to represent the ‘playbook’ of best responses, where the transformer implicitly selects the appropriate best response at test time, while  PLASTIC uses an explicit Bayesian teammate selection module.


    **Minor suggestions**

    - CooT architecture is not described. Since the use of the transformer architecture is the main novelty, it should be described somewhere.


**Citations**

[1] Ravula, M., Alkoby, S., & Stone, P. (2019). Ad Hoc Teamwork With Behavior Switching Agents. *Proceedings of the Twenty-Eighth International Joint Conference on Artificial Intelligence*, 550–556. [https://doi.org/10.24963/ijcai.2019/78](https://doi.org/10.24963/ijcai.2019/78)

[2] Zhang, Z., Yuan, L., Li, L., Xue, K., Jia, C., Guan, C., Qian, C., & Yu, Y. (2023). Fast Teammate Adaptation in the Presence of Sudden Policy Change. *Proceedings of the Thirty-Ninth Conference on Uncertainty in Artificial Intelligence*, 2465–2476. [https://proceedings.mlr.press/v216/zhang23a.html](https://proceedings.mlr.press/v216/zhang23a.html)

[3] Weis, M. A., Wołczyk, M., Nasser, R., Saurous, R. A., Arcas, B. A. y, Sacramento, J., & Meulemans, A. (2026). *Multi-agent cooperation through in-context co-player inference* (arXiv:2602.16301). arXiv. [https://doi.org/10.48550/arXiv.2602.16301](https://doi.org/10.48550/arXiv.2602.16301)

[4] Zhang, X., Chan, H., Ye, D., Cai, Y., & Zhao, M. (2025, June 18). *Ad Hoc Teamwork via Offline Goal-Based Decision Transformers*. Forty-second International Conference on Machine Learning. [https://openreview.net/forum?id=tl3FlgWScA&noteId=OK5Hi25wOc](https://openreview.net/forum?id=tl3FlgWScA&noteId=OK5Hi25wOc)

[5] Mon-Williams, R., Taylor-Davies, M., Mieczkowski, E., Velez, N., Bramley, N. R., Wang, Y., Griffiths, T. L., & Lucas, C. G. (2025). *Partner Modelling Emerges in Recurrent Agents (But Only When It Matters)* (arXiv:2505.17323). arXiv. [https://doi.org/10.48550/arXiv.2505.17323](https://doi.org/10.48550/arXiv.2505.17323)

[6] Barrett, S., Rosenfeld, A., Kraus, S., & Stone, P. (2017). Making friends on the fly: Cooperating with new teammates. *Artificial Intelligence*, *242*, 132–171. [https://doi.org/10.1016/j.artint.2016.10.005](https://doi.org/10.1016/j.artint.2016.10.005)

---

> ### Author Rebuttal · Authors · 2026-03-31
>
> We sincerely thank the reviewer for the thorough and constructive comments.
>
> > The method appears to amount to doing BC with a transformer policy
>
> We would like to clarify: our method is not “doing BC with a transformer policy” conditioning on within-episode history; instead, our method uses cross-episode best-response trajectories as context, with query states sampled independently (see “query state sampling strategy” to Reviewer AQ11). This design enables partner-specific adaptation (see Figure 2).
>
> > Transformers have been studied for decision-making
>
> [3] studies in-context co-player inference in social dilemmas with mixed motives, while our work focuses on adapting to diverse partner conventions under shared rewards in a pure cooperative setup.
>
> [4] requires explicit teammate-aware subgoal supervision; yet, our method conditions on raw cross-episode interaction history without such supervision. Crucially, [4]  is designed for the offline setting and does not perform test-time adaptation, while CooT can adapt to unseen partners without any parameter updates via context update.
>
> [5] shows that partner modeling can emerge in recurrent agents under certain conditions. On the other hand, our work designs a practical framework that can in-context learn and adapt to partner behaviors at test time through implicitly modeling the partner.
>
> We will revise the paper to include the above discussion.
>
> > As the number of agents scales, the context length will grow linearly
>
> We would like to clarify that CooT’s context does not scale as the number of agents grows, as the context consists solely of trajectories from the best response.
>
> > GRF only 3 agents
>
> We used GRF to test higher-dimensional observations and longer horizons. While we agree that scaling from 2 agents in Overcooked to 3 agents in GRF is limited, we still believe this step is meaningful. We will revise the paper to tone down the scalability claim.
>
> > Comparison to recurrent BC
>
> As suggested by the reviewer, we conducted an experiment with BC+RNN. The results below show that BC+RNN provides only marginal improvements over BC, and sometimes underperforms. This supports our intuition: recurrence could potentially help with modeling temporal dependencies, but does not address the core challenge of generalizing to unseen partners. We will revise the paper to include this result.
>
> |Method|Coord. Ring|Coord. Ring Multi-recipe|Counter Circ.|Asymm. Adv.|Bothway Coord.|Overall|
> |:-:|:-:|:-:|:-:|:-:|:-:|:-:|
> |BC|26.2±1.8|9.0±0.5|10.8±5.3|108.8±6.1|99.0±1.3|50.8±2.0|
> |BC-RNN|28.1±1.7|22.0±0.2|15.2±0.8|105.5±4.2|93.6±1.0|52.9 ±1.0|
> |CooT (ours)|**38.3±3.7**|**46.0±4.0**|**28.3±2.3**|**129.5±9.3**|**101.9±1.0**|**68.8±2.3**|
>
> > Fairness of population-based baselines
>
> The comparison to MEP and HSP is proper and fair. HSP is precisely the baseline the reviewer asks for, a population-based RL policy trained against all 36 training partners. Our method CooT learns from the same partner distribution. Hence, the comparison between CooT and HSP is fair. Since MEP generates training partners by maximizing population entropy, evaluating it under an external population (e.g., from HSP) would contradict its design. Thus, we only used the 15 MEP-generated agents.
>
> > There should be a comparison against PLASTIC
>
> PLASTIC [6] assumes test-time partners to be similar to training partners, and selects among a fixed set of best-response policies via Bayesian belief updates. In contrast, CooT models partner implicitly and can generalize/interpolate beyond training best responses, through in-context adaptation.
>
> We implemented PLASTIC in Coord. Ring and report the result below, which shows that retrieving from training best-response policies may be insufficient when test-time partners exhibit different behaviors than training, as CooT can adapt better.
>
> |Method|Coord. Ring|
> |:-:|:-:|
> |PLASTIC|19.4±0.4|
> |CooT (ours)|**38.3±3.7**|
>
> We will revise the paper to discuss PLASTIC.
>
> > CooT architecture is not described
>
> The architecture is presented in Table 9, including \# of hidden layers and attention heads, and embedding size.
>
> > Two possible limitations: compounding error & requirements for diversity partners
>
> While CooT utilizes a supervised objective, it is fundamentally distinct from imitation learning methods like BC, which could suffer from compounding error. CooT performs in-context learning across episodes and updates its belief about the partner’s policy, rather than mimicking a fixed trajectory (which drifts). This “belief tracking” allows the agent to reduce uncertainty rather than compound errors [7]. This is empirically confirmed by Figure 2, which shows that CooT's performance improves over episodes.
>
> We acknowledge that our method requires a diverse partner dataset, as do all population-based methods, including HSP, MEP, PACE, etc.
>
> **References**
>
> [1-6] follow the review
>
> [7] Lee et al. “Supervised pretraining can learn in-context reinforcement learning.” NeurIPS, 2023.

---

> > ### Author Rebuttal · Reviewer_Cbxd · 2026-04-03
> >
> > Thank you for providing the above discussion; it has improved my understanding of the paper and resolved some of my concerns.
> >
> > I agree now that COOT is not the same as recurrent BC, which was one of my previous concerns. My main point of confusion was the query state, which I did not realize was sampled independently from the context. I believe this could be better explained in the paper, perhaps by modifying Figure 1a to show a separate dataset box for the query state.
> >
> > I have a few follow-up questions and comments.
> >
> > 1. While COOT is not equivalent to recurrent BC, it is still trained using a maximum likelihood loss on a dataset of "optimal" behavior (in the sense that the actions are provided by the best responses). The main difference seems to be the context from the offline dataset of partners and best response trajectories. Thus, it still seems similar to BC. Can the authors further clarify the relationship between BC and COOT, focusing on why there is no compounding error? I was not convinced by the explanation that COOT performs belief-tracking rather than mimicking fixed trajectories. For COOT, what happens if it encounters a query state that was not present in the training dataset? Why would there not be error in the predicted action?
> > 2. Is the query state simply appended to the context? Or does it function as the query within the attention layers? This was ambiguous to me.
> > 3. I strongly suggest that the authors add that COOT uses a GPT-2 architecture to the main paper. This provides important context for understanding the method. Table 9 contains hyperparameters, which doesn't seem like the right place for the architecture, which is a crucial aspect of their method.
> > 4. What was the architecture used for BC? I did not find it in Table 9.
> > 5. On HSP:
> > > HSP is precisely the baseline the reviewer asks for, a population-based RL policy trained against all 36 training partners.
> >
> >  The HSP method does not consist of training an AHT agent against any population of policies. It prescribes a specific way to generate the partners, based on varying a feature-based reward function to generate diverse behaviors followed by a filtering step. Upon a reread of Section 4.2 combined with the authors' comment, I realized that COOT leverages the exact same dataset generation pipeline as HSP. It should be clarified in the experiments section that HSP and COOT rely on the same partner pool. The discussion of related work should also acknowledge this relationship.

---

> > > ### Author Response · Authors · 2026-04-07
> > >
> > > We sincerely thank the reviewer for acknowledging the rebuttal and for the follow-up questions. Please find the response to the questions below.
> > >
> > > > Relationship between BC and COOT, focusing on why there is no compounding error.
> > >
> > > We acknowledge that our previous response was unclear. Please find the revised explanation below.
> > >
> > > The key distinction from BC lies in the training objective. BC learns a fixed mapping from states to actions along sequential trajectories. CooT instead learns a mapping $(\text{context}, s_t) \rightarrow a_t$, where the query state $s_t$ is sampled independently from the dataset during training, decoupled from the context trajectories. This trains the model to jointly condition on both the current state and context, reducing the relative importance of the query state alone. Therefore, the model can still leverage context to infer partner behavior and take a reasonable action when a previous action was suboptimal, preventing errors from compounding.
> > >
> > > More importantly, while intra-episode compounding errors may still occur, CooT's context conditioning enables posterior sampling across episodes (Theorem 1 in [1]): each new episode adds interactions to the context, providing more information about the partner's behavior. This information gain grows linearly with the number of episodes $O(K)$ in the context, where $K$ denotes the number of episodes, while the cumulative cost of imperfect predictions only grows as $\tilde{O}(\sqrt{K})$ (Corollary 6.2 in [1]). Since information accumulates faster than errors do, the average per-episode regret decreases as $\tilde{O}(1/\sqrt{K})$, which means CooT gets closer to optimal behavior as more episodes are observed. This is the opposite of compounding error, where mistakes accumulate faster than the agent can correct them. This is supported by Figure 2, which shows CooT's performance steadily improving over episodes.
> > >
> > > We thank the reviewer for inspiring us to improve our submission. We will include this discussion in the revised paper.
> > >
> > > [1] Lee et al. “Supervised pretraining can learn in-context reinforcement learning.” NeurIPS, 2023.
> > >
> > > > On prediction errors under unseen query states.
> > >
> > > When CooT encounters an unseen query state, prediction errors can occur, as with any supervised method. However, unlike BC, which relies solely on the query state, CooT also conditions on context. This context captures the partner’s behavioral pattern and provides an additional signal for action selection, enabling more consistent behavior when the state alone is insufficient.
> > >
> > > > Is the query state simply appended to the context?
> > >
> > > Yes. The query state is appended as the final token, i.e., the input sequence is
> > > $[\tau\_1, \tau\_2, \ldots, \tau\_T, s\_{\text{query}}]$. Each context transition $(s\_i, a\_i, r\_i)$ and the query state $s\_{\text{query}}$ (with zero-padded action and reward) are projected into a shared embedding space via a common linear layer.
> > >
> > > The predicted action is read from the hidden state of the final token (the query). Through attention, this token accesses all context tokens, enabling implicit inference of partner behavior from the interaction history.
> > >
> > > > Model architecture placement
> > >
> > > We agree with the reviewer and will revise the paper to clearly describe the architecture in Section 4.
> > >
> > > > BC architecture.
> > >
> > > The BC baseline uses a 2-layer MLP with a hidden size of 32 to map a state to an action. We will include this in Table 9 in the revised paper.
> > >
> > > > On the shared partner generation pipeline with HSP and clarifying HSP.
> > >
> > > We will revise the paper to clearly state that CooT leverages the same dataset generation pipeline as HSP and therefore shares the same partner pool. We will also update the experiments section and related work to explicitly acknowledge this relationship.
> > >
> > >
> > > HSP prescribes a method for generating diverse partners via reward variation, and then trains an RL policy to generalize across them. The key distinction lies in how this population is used: whereas HSP trains an RL policy over this population, CooT uses it to construct supervised training data and learns a context-conditioned model that predicts partner-specific best-response actions from interaction history.
> > >
> > > # Final remark
> > >
> > > As this is the last round of responses we are allowed to post, we would like to take this opportunity to sincerely thank the reviewer for the thoughtful and constructive review and for engaging promptly with our rebuttal, providing detailed follow-up questions. We genuinely appreciate the time and care you put into the review, which helps us to clarify, reflect, and improve our work. We will incorporate all the discussion in the revised paper.

---

### Official Review · Reviewer_AQ11 · 2026-03-13

**Soundness:** 3
**Presentation:** 3
**Significance:** 3
**Originality:** 3
**Overall Recommendation:** 4
**Confidence:** 3

**Summary:**

This paper proposes Coordination Transformers, an in-context learning framework for real-time adaptation to unseen partners in cooperative multi-agent settings. The method conditions action prediction on recent interaction histories and aims to improve few-shot coordination without test-time parameter updates.

**Compliance With Llm Reviewing Policy:**

Affirmed.

**Final Justification:**

I am satisfied with the authors’ additional comparison with GAMMA and will keep my score unchanged.

**Key Questions For Authors:**

1. **How are the numbers in Table 1 computed exactly?**
   Since COOT adapts across episodes by appending completed trajectories to context, please clarify whether the reported results are averaged over 50 consecutive episodes per partner, taken from a fixed episode index, or aggregated in some other way.

2. **How are query states sampled in Section 4.2?**
   The paper states that query states are sampled from each context. Is this random sampling, and if so, how sensitive is performance to the sampling procedure?

3. **Can you include oracle best-response returns in Table 1?**
   Since BR-prox is normalized by the best-response return for each partner, reporting the absolute best-response/oracle return would make the results easier to interpret.

**Limitations:**

yes. The paper explicitly acknowledges a limitation of the current approach, namely that it relies solely on actions and does not incorporate explicit communication between partners

**Strengths And Weaknesses:**

**Strengths:**
* Using in-context learning for partner adaptation in cooperative MARL is interesting and appears novel. The submission aims to discuss an important concept: whether full interaction histories can support rapid coordination with unseen partners without online fine-tuning.
* The results are generally strong across Overcooked and GRF, and the adaptation-over-episodes analysis is a useful addition.
* The human evaluation is a valuable part of the paper and strengthens the practical relevance of the work.

**Weaknesses:**
* The evaluation protocol in Table 1 is not precise enough. Since COOT explicitly adapts across episodes by appending completed trajectories to context, it is important to clarify exactly how the reported numbers are computed: e.g., averaged over 50 consecutive episodes, over a fixed episode index, or aggregated differently across partners. This materially affects interpretation and fairness.
*  The table 1 reports average return and BR-prox, but not the absolute partner-specific best-response return. Including oracle returns would make it easier to assess how far each method remains from the coordination optimum in absolute terms.
* The paper states “From each context, we sample query states$ \(s_t\)$,” but it is unclear how this sampling is done and whether the choice affects performance.
* A baseline such as [1]  seems relevant, especially because it would be informative to compare first-episode performance in a zero-coordination setting.
* It would be helpful to include a dataset-scale ablation, for example varying the number of partner pairs, contexts, or total training samples, to understand how COOT scales with data and whether its gains persist in smaller-data regimes.

[1]  Liang et al,"Learning to Cooperate with Humans using Generative Agents," NeurIPS 2024

---

> ### Author Rebuttal · Authors · 2026-03-31
>
> We sincerely thank the reviewer for the thorough and constructive comments. Please find the response to your questions below.
>
> > Evaluation protocol: how the reported numbers are computed
>
> For each evaluation partner, we run 50 consecutive episodes, where CooT continuously adapts by appending trajectories to its context across episodes. We then average these results across all evaluation partners, and finally report the mean over 3 random training seeds.
> We use 10 evaluation partners for all Overcooked layouts and GRF, except for Coord. Ring Multi-recipe, where we use 15 partners due to higher strategy diversity. We will revise the caption of Table 1 to make this protocol explicit.
>
> > Absolute partner-specific BR return
>
> As requested by the reviewer, we report the **absolute partner-specific best-response return** of our method CooT for each layout below.
>
> |Evaluation Partner \#|Bothway Coord.|Coord. Ring|Coord. Ring Multi-Recipe|Counter Circuit|Asymm. Adv|
> |:-:|:-:|:-:|:-:|:-:|:-:|
> |1|100|80|110|120|220|
> |2|140|120|90|80|160|
> |3|100|100|100|60|220|
> |4|160|100|80|100|220|
> |5|0|140|90|80|220|
> |6|100|100|100|60|0|
> |7|60|100|90|80|220|
> |8|60|40|80|120|80|
> |9|140|80|90|80|200|
> |10|100|0|100|120|200|
> |11|||110|||
> |12|||100|||
> |13|||90|||
> |14|||70|||
> |15|||90|||
>
> We will revise the appendix to include these numbers.
> > Query state sampling strategy
>
> During training, we sample the query state $s_t$ **independently of the context trajectories**, uniformly from the full offline dataset $\mathcal{D}$ (i.e., $s_t \sim \mathcal{D}$). Intuitively, this design prevents the model from relying on simple trajectory continuation that can arise when $s_t$ is tied to the temporal sequence in the context. If $s_t$ were restricted to the context window, the model could solve the task by continuing trajectories rather than inferring partner behavior. Decoupling $s_t$ instead forces the model to use interaction history, enabling effective adaptation to unseen partners at test time. We will revise Section 4.2 and Appendix B.2 to clarify them.
>
> > “Learning to Cooperate with Humans using Generative Agents” (GAMMA)
>
> We thank the reviewer for highlighting GAMMA, which makes excellent contributions to Ad-Hoc Teamwork.
>
> GAMMA focuses on enriching the distribution of training partners; it uses a VAE to generate a continuous space of diverse partner policies for an RL agent to train against. In contrast, our work focuses on the agent architecture, using Transformers to enable rapid, in-context adaptation to a partner's behavior during the episode.
>
> Since GAMMA and our work tackle fundamentally different problems (partner diversity vs. test-time adaptation), it’s difficult to make a direct comparison. We will revise the paper to include a discussion of GAMMA.
>
> > Dataset-Scale Ablation
>
> Following the reviewer’s suggestion, we performed ablations on (i) the number of training partners, and (ii) the number of trajectories collected per training partner on the Coord. Ring layout,
>
> **Experiment 1: Scaling down the number of training partners: 36 (original)→24→12**
> |Method|36|24|12|
> |:-:|:-:|:-:|:-:|
> |BC|26.24 ± 1.80|21.47 ± 0.31|15.34 ± 1.28|
> |CooT (ours)|38.30 ± 3.71|33.66 ± 4.03|29.11 ± 1.16|
>
> Our method CooT is more robust to reduced partner diversity: performance drops by 24.0% (36→12), compared to 41.5% for BC.
>
> **Experiment 2: Scaling down the number of trajectories collected per training partner: 250 (original)→200→150**
> |Method|250|200|150|
> |:-:|:-:|:-:|:-:|
> |BC|26.24 ± 1.80|23.70 ± 0.96|17.81 ± 1.33|
> |CooT (ours)|38.30 ± 3.71|29.72 ± 3.05|27.35 ± 1.03|
>
> CooT also degrades more gracefully with fewer episodes (28.6% drop vs. 32.1% for BC).
>
> We thank the reviewer for the insightful suggestion and will include these results in the revised paper.
>
> For context length ablation, we note that this analysis is already reported in Table 12 of our submission.

---

> > ### Author Rebuttal · Reviewer_AQ11 · 2026-04-03
> >
> > I agree that GAMMA and CooT do not target exactly the same problem: GAMMA focuses more on zero-shot coordination through partner diversity during training, whereas CooT focuses on test-time adaptation through accumulated interaction context. However, both methods are ultimately concerned with adaptation to previously unseen partners in cooperative settings. For this reason, I do not think the comparison is uninformative. If anything, GAMMA operates in a stricter zero-shot setting, so comparing first-episode or early-episode performance would still be meaningful and would help position CooT more clearly relative to prior work on ad-hoc coordination.

---

> > > ### Author Response · Authors · 2026-04-08
> > >
> > > We sincerely thank the reviewers for the thoughtful review and for engaging promptly with our rebuttal with positive acknowledgment. We really appreciate the time and care you put into our work.
> > >
> > > > I agree that GAMMA and CooT do not target exactly the same problem: GAMMA focuses more on zero-shot coordination through partner diversity during training, whereas CooT focuses on test-time adaptation through accumulated interaction context. However, both methods are ultimately concerned with adaptation to previously unseen partners in cooperative settings. For this reason, I do not think the comparison is uninformative. If anything, GAMMA operates in a stricter zero-shot setting, so comparing first-episode or early-episode performance would still be meaningful and would help position CooT more clearly relative to prior work on ad-hoc coordination.
> > >
> > > We thank the reviewer for the suggestion, and we have now implemented GAMMA as an additional baseline on both Coord. Ring and Counter Circ.
> > >
> > > Following the original setup, we trained GAMMA adaptive agents to converge on each layout and evaluated them using the same partner pool as in our experiments. To reflect GAMMA’s focus on zero-shot coordination, we report performance at episode 1 (*Ep 1 (zero-shot)*), averaged over episodes 1-5 (*Mean Return (Ep1–5)*), and averaged over episodes 1-10 (*Mean Return (Ep1–10)*) to capture the few-shot regime. All the results below are averaged over 3 random seeds (mean ± std).
> > >
> > > **Coord. Ring**
> > > |Method|Ep1 (Zero-shot)|Mean Return (Ep1–5)|Mean Return (Ep1–10)|
> > > |--|--|--|--|
> > > |GAMMA|**29.80±6.07**|-|-|
> > > |CooT|**29.63±4.63**|39.56±6.46|41.11±5.23|
> > >
> > > [Adaptation performance for Coord. Ring](https://drive.google.com/file/d/1R5Vk6rRFLBshS62eyglw9UrdcXWDARHz/view?usp=drive_link)
> > >
> > > **Counter Circ.**
> > > |Method|Ep1 (Zero-shot)|Mean Return (Ep1–5)|Mean Return (Ep1–10)|
> > > |--|--|--|--|
> > > |GAMMA|9.60±2.42|-|-|
> > > |CooT|**21.33±6.11**|21.07±5.69|22.73±3.40|
> > >
> > > [Adaptation performance for Counter Circ.](https://drive.google.com/file/d/1C9FKfBoqtadbA6fuG6Le8XOgtDqfTPX0/view?usp=drive_link)
> > >
> > > These results show that CooT matches or exceeds GAMMA in the zero-shot setting (comparable on Coord. Ring and higher on Counter Circ.), while further improving with additional interaction through in-context adaptation. This is consistent with CooT’s training: it is exposed to diverse context conditions, including empty-context cases, enabling strong zero-shot performance without sacrificing its ability to adapt online.
> > >
> > > We follow [the official GAMMA implementation](https://github.com/lych1233/GAMMA-human-ai-collaboration) and training pipeline, using the recommended VAEs (best_logp_kl_32 for Counter Circ. and kl_32 for Coord. Ring) trained on trajectories from a population of 32 agents (24 MEP + 4 BC + 6 held-out). The only modification is setting the episode length to 200 (vs. 400 in the original paper) to match our environment. As a result, the primary difference lies in the evaluation partners.
> > >
> > > This difference is important: the original GAMMA evaluation uses human partners who adapt to the agent, whereas our main evaluation partners have fixed latent preferences and do not react to the agent’s behavior. This creates a more challenging and diverse coordination setting—especially in Counter Circ., which has a more multimodal strategy space. In this regime, CooT’s explicit conditioning on interaction history allows it to infer and adapt to each partner more effectively.
> > >
> > > We believe this comparison is fair and directly comparable, as both GAMMA and CooT are evaluated against the same partner pool (and GAMMA originally evaluates its VAE-trained adaptive agent against an MEP adaptive agent trained with a population of 36 agents — the same setup as our MEP baseline).
> > >
> > > We thank the reviewer for the suggestion. We will revise the paper to discuss this result.
> > >
> > > We hope these additional experiments fully address the concern, and the reviewer would consider updating the score in light of this new results.

---

### Official Review · Reviewer_syy1 · 2026-04-02

**Soundness:** 2
**Presentation:** 2
**Significance:** 3
**Originality:** 3
**Overall Recommendation:** 3
**Confidence:** 4

**Summary:**

This paper proposes COOT, a multi-agent collaboration framework based on contextual learning. Its core idea is to transform partner adaptation problems into sequence modeling. In the training phase, interaction trajectories between behavior-preference agents and their optimal responses are collected. In the testing phase, a Transformer is trained to predict optimal response actions by observing the partner's past behavior, without requiring parameter updates. Experiments comparing COOT to multiple baselines in Overcooked and Google Research Football, along with human user surveys, demonstrate that COOT offers significant advantages in complex collaboration scenarios.

**Compliance With Llm Reviewing Policy:**

Affirmed.

**Ethical Review Concerns:**

This paper involves Human Study.

**Ethical Review Flag:**

Flag this paper for an ethics review.

**Final Justification:**

COOT's core idea is innovative, and its experimental design is relatively solid; the human research aspect is a plus. However, the overlap between training and testing distributions, the lack of sufficient analysis of some baseline anomalies, and the fact that the method's advantages are mainly concentrated in complex scenarios all require more serious discussion. The current version's argument is close to, but does not yet reach, the standards of a top-tier conference.

**Key Questions For Authors:**

see weaknesses

**Limitations:**

see weaknesses

**Strengths And Weaknesses:**

Strengths

The approach of reframing the partner adaptation problem as an in-context learning problem is innovative. Unlike existing meta-reinforcement learning methods that rely on compressed latent representations, the design that directly conditions the complete interaction history is intuitive, rational, and experimentally supported.

The experiments are comprehensive, encompassing multiple layouts, various baseline types (population-based, fine-tuning, meta-reinforcement learning, context-based), and generalization tests from reinforcement learning agents to actual human partners, making them highly compelling overall.

The human user survey design adheres to high standards, being double-blind, IRB-reviewed, involving 36 participants, and combining quantitative and qualitative analysis.

Weaknesses:

COOT's training data is essentially supervised learning of optimal response trajectories, and the unknown partners that appear during testing are also generated from the same event-based reward distribution. Because the degree of overlap between the training and test data distributions has not been systematically analyzed, the generalization ability of this method to partners completely outside the distribution (e.g., human-trained agents or other algorithms with entirely different behavioral logic) remains questionable. The current scale of human experiments (36 participants, single layout) is insufficient to fully answer this question.

COOT slightly underperforms HSP in the Asymmetry Advantage (Asymm. Adv.) and performs similarly to HSP in the Coord. Ring (Coord. Ring), showing a significant advantage only in more complex layouts. This pattern suggests that COOT's superiority may not be due to its inherently stronger general-purpose cooperative capabilities, but rather primarily to the failures of other methods in complex scenarios; however, this point is not sufficiently discussed in the paper.

HSP-meta and PACE showed surprisingly low performance in multiple layouts, even inferior to standard HSP. The explanations presented in the paper (noisy latent variables, misaligned reconstruction loss, etc.) are post-hoc analyses and lack validation through ablation studies. Furthermore, it is unclear whether these two baselines were properly hyperparameter-tuned, affecting the fairness of the comparison.

The implicit assumption of this method is that partner behavior can be sufficiently identified from a limited number of interaction histories. However, the sensitivity analysis to context length (Table 12) is performed only in a single layout, and the reason for the performance drop at 7 episodes is not explained. Additionally, the computational cost of COOT (approximately 19 hours of training per layout) and its reliance on pre-trained optimal response data are not discussed in the paper regarding their feasibility in practical applications.

---

### Decision · Program_Chairs · 2026-04-30

**Decision:**

Accept (regular)

**Comment:**

The paper considers a multi-agent coordination problem, where the goal is to efficiently coordinate with unseen partners. The paper proposes a novel approach based on in-context learning (CooT) and evaluates it using two existing benchmarks (Overcooked and Google Research Football) and a human subject study. CooT outperforms baselines and was more favorable by the human participants in the human subject study. Applying in-context learning in the considered problem setting appears novel, and the paper provides rather extensive experiments to showcase the utility of in-context learning for partner adaptation in cooperative MARL. That said, the approach has a somewhat limited methodological novelty, and several concerns have been raised about the evaluation. While some of these concerns were addressed by the rebuttal, the paper could benefit from adding additional baselines, and expanding the ablation and human subject studies. The authors are encouraged to revise their paper accordingly.